# Mapping landscape connectivity as a driver of species richness under tectonic and climatic forcing

Tristan Salles[1], Patrice Rey[1], and Enrico Bertuzzo[2]

[1]School of Geosciences, University of Sydney, Sydney, NSW, 2006, Australia
[2]Dipartimento di Scienze Ambientali, Informatica e Statistica, Università Ca'Foscari Venezia, Venice, Italy

**Correspondence:** Tristan Salles (tristan.salles@sydney.edu.au)

**Abstract.** Species distribution and richness ultimately result from complex interactions between biological, physical and environmental factors. It has been recently shown for a static natural landscape that the elevational connectivity, which measures the proximity of a site to others with similar habitats, is a key physical driver of local species richness. Here we examine changes in elevational connectivity during mountain building using a landscape evolution model. We find that under uniform tectonic and variable climatic forcing, connectivity peaks at mid-elevations when the landscape reaches its geomorphic steady-state and that the orographic effect on geomorphic evolution tends to favour lower connectivity on leeward facing catchments. Statistical comparisons between connectivity distribution and results from a metacommunity model confirm that to the first order, landscape elevation connectivity explains species richness in simulated mountainous regions. Our results also predict that low connectivity areas which favour isolation, a driver for in-situ speciation, are distributed across the entire elevational range for simulated orogenic cycles. Adjustments of catchment morphology after cessation of tectonic activity should reduce speciation by decreasing the number of isolated regions.

## 1 Introduction

The idea that mountainous landscapes play a role in biological evolution has a long history that can be tracked back to Darwin and Wallace, when fauna boundaries were noted to correspond to physiographic discontinuities and gradients (Wallace, 1860). Over geological time scales (millions of years), surface processes including erosion and incision conspire to undo surface uplift driven by tectonic and geodynamic processes. These competing processes can convert landscapes of low elevation and relief, homogeneous environment and low resistance to migration, into complex landscapes with sharp environmental gradients, and fragmented habitats separated by migratory corridors (Steinbauer et al., 2016).

Several studies have shown that on geological timescales, changes in landscape morphology stimulates migratory behaviour, dispersal and redistribution as species track their optimum habitats (Gillespie and Roderick, 2014; Craw et al., 2015; Muneepeerakul et al., 2019). Species running out of favourable habitats are forced to coexist with other ones and adapt, which leads to

speciation, increasing endemism and biodiversity (Smith et al., 2014). As the mountainous landscape becomes more complex and diverse, environmental gradients increase and species have to move across shorter distances to track suitable habitats (Smith et al., 2014; Elsen and Tingley, 2015) and find refuges to survive both long-term mountain building processes and short-term climatic changes. Hence, mountains host a disproportionately large fraction of terrestrial species (Badgley, 2010; Hoorn et al., 2010; Steinbauer et al., 2018), illustrating the tectonic influence on ecology and evolutionary biology.

Determining the underlying geophysical drivers of species richness requires accounting for factors (gradients of topography, habitat capacity, humidity, temperature, altitude, solar exposure...) that often covary with elevation (Lomolino, 2001; Kearney and Porter, 2009; Martensen et al., 2017; Liu et al., 2018). Recently, Bertuzzo et al. (2016) proposed a metric to assess biodiversity patterns. Such metric, called the landscape elevational connectivity (LEC), accounts for two fundamental landscape geomorphological properties: (1) within mountains ranges the maximum surface area peaks for mid-elevations rather than lowest elevations, and (2) at mid-elevations species have the option to disperse up or down to well connected suitable patches. As a consequence of these two properties, species richness (*e.g.*, diversity) is predicted to be highest at mid-elevations, showing a hump-shaped pattern (Bertuzzo et al., 2016; Lomolino, 2001; Rahbek, 1995, 2005; McCain and Grytnes, 2010; Kessler et al., 2011) because diversity is promoted by increased area and connectivity (MacArthur and Wilson, 1967; Solé and Bascompte, 2006; Economo and Keitt, 2010; Carrara et al., 2012).

The results from Bertuzzo et al. (2016) were obtained on a static landscape. The aim of this study is to extent the analysis over the entire orogenic cycle from mountain building to relaxation phases using a synthetic example and to map over space and time the landscape connectivity. First we analyse temporal and spatial LEC distribution based on the results of a landscape elevation model (Salles, 2016) under uniform tectonic and variable climatic conditions. Then, we run a zero-sum metacommunity model (Hubbell, 2001) on the simulated synthetic surfaces and, as in Bertuzzo et al. (2016), we find that the LEC quantity explains to the first order species richness in mountainous regions and can be used to infer changing patterns of biodiversity resulting from the effect of climate, tectonic and geomorphic processes. The obtained results provide new insights on how the tempo imposed by landscape morphological changes over geological time scales affects the distribution of species richness in mountainous regions.

## 2 Methods and experimental design

### 2.1 Coupled landscape evolution and orographic precipitation model

In this paper, the numerical simulations of mountain evolution are performed with *Badlands* landscape evolution model (Salles, 2016; Salles et al., 2018a) and fluvial erosion rates as well as predicted sediment transport in rivers are solved using the stream-power law (SPL). The SPL relates the erosion rate $I$ to the product of mean annual net precipitation rate ($\bar{P}$), drainage area ($A$) and local slope ($S$) and takes the form:

$$I = \kappa_e(\bar{P}A)^m S^n \tag{1}$$

where the erodibility coefficient $\kappa_e$ is controlled by climate and lithology, $m$ and $n$ are positive exponents (Chen et al., 2014) that mostly depend on the nature of the dominant erosional mechanism (Dietrich et al., 1995). Regardless of its simplicity, the SPL (Eq. 1) allows us to simulate the main geomorphic characteristics of mountainous regions, in which landscape evolution is dominated by detachment-limited erosion regime (Whipple and Tucker, 1999).

In addition to riverine processes, soil creep (approximated by a diffusion law) is used to account for semi-continuous processes of soil displacement (Tucker and Hancock, 2010; Salles et al., 2018a):

$$D = \kappa_d \nabla^2 z \tag{2}$$

$z$ is the local landscape elevation and $\kappa_d$ is the diffusion coefficient. This formulation assumes a linear dependency of superficial sediment transport with topographic gradient.

Accounting for the two processes defined above, the equation of mass conservation driving landscape temporal evolution is expressed as:

$$\frac{\partial z}{\partial t} = U - I + D \tag{3}$$

where $t$ is time and $U$ is the rock uplift rate. It is worth noting that the landscape evolution scenarios presented in this study neglect spatial and temporal variations in tectonic evolution and rock strength, as well as the influence of sediment characteristics

on river incision.

Precipitation patterns are known to impact geomorphic evolution at local (ridge-valley) to regional (full mountain range) scales (Anders et al., 2008; Bonnet, 2009). To estimate the controls of precipitation variability on topography evolution and on associated landscape connectivity, the set of landscape evolution equations presented above are coupled with a linear orographic precipitation model (Smith and Barstad, 2004). The approach already implemented in Badlands (Salles et al., 2018b) only

accounts for the first-order physics of orographic precipitation and computes rainfall under idealised climatic conditions defined by a specific wind velocity and direction (Anders et al., 2008) and assumes steady, uniform, and saturated air flow (Fig. 1a).

In the study, analysis of catchment dynamics are performed using river longitudinal profiles as well as $\chi$ plots and maps. $\chi$ analysis is a method of extracting information from channel profiles that attempts to compare channels with different discharges (Perron and Royden, 2013). The longitudinal coordinate $\chi$ has dimensions of length and is linearly related to the elevation

(Mudd et al., 2014). The quantity $\chi$ at any channel point depends on the river network geometry and estimates the dynamic state of river basins (Willett et al., 2014):

$$\chi = \int_{x_b}^{x} \left( \frac{A_0}{A(x')} \right)^{m/n} dx' \tag{4}$$

with $x$ the distance along a given channel channel from the river base (the outlet) $x_b$ and the point considered, $A_0$ a scaling area, $m$ and $n$ the coefficient of the SPL (Eq. 1) and $A(x')$ the upstream drainage area at position $x'$. Mapping $\chi$ along channel

networks and comparing values of $\chi$ across drainage divides provide a measure of the equilibrium state of a river network under the influence of tectonic uplift and river erosion.

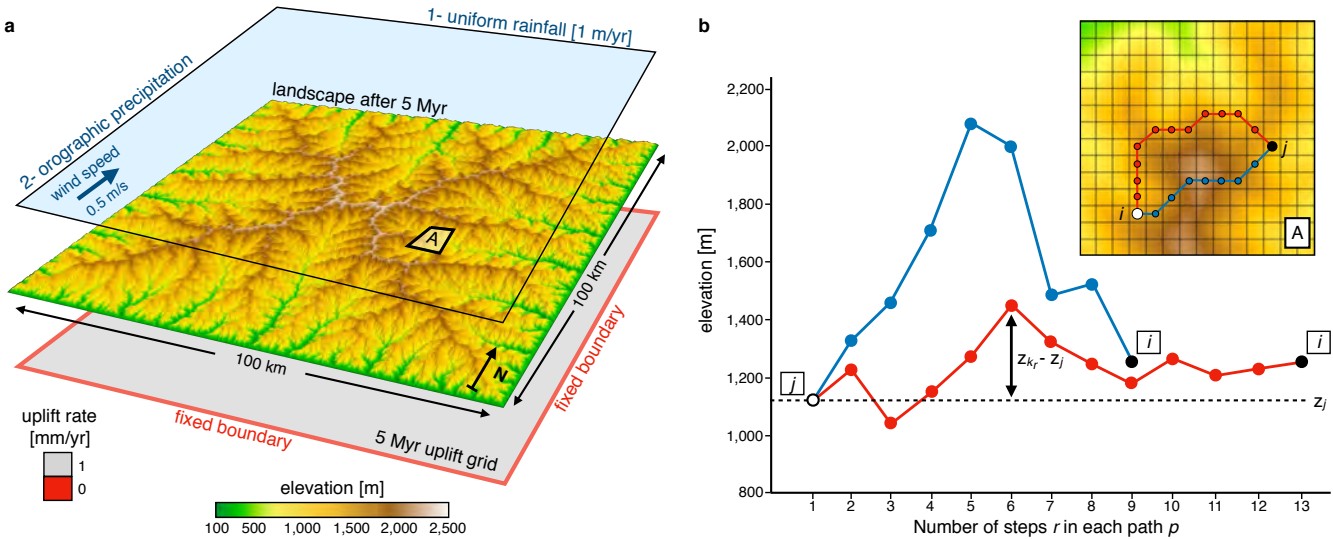

**Figure 1. a)** Experimental design showing modelled landscape after 5 Myr and the controlling forcing conditions. Two scenarios are ran. In the first one a constant precipitation (1 m/yr) is applied, then in the second, an orographic precipitation model is used with constant wind speed (0.5 m/s) coming from the South. For both scenarios, the model is forced with an initial uplift phase (uniform rate of 1 mm/yr) through the first 5 Myr. **b)** Approach used to compute the closeness measure ($C_{ji}$) required to quantify landscape elevational connectivity (LEC) based on the topography grid (*e.g.*,region A) defined in panel a). Two possible paths connecting site $j$ to $i$ (inset) are proposed with their corresponding elevation profiles. Associated costs are computed following Eq. 6 ($\sum_{r=2}^{L}(z_{k_r} - z_j)^2$). Despite a longer length, the cost associated to the red path is smaller than that of the blue one as it travels through sites with elevations more similar to $z_j$ (adapted from Bertuzzo et al. (2016)).

## 2.2 Landscape elevational connectivity

Landscape connectivity encapsulates the combined effects of (1) landscape morphological structure and (2) the species ability to move outside its usual niche width (Tischendorf and Fahrig, 2000). As such it is both species- and landscape-specific. Here, we generalise the concept and assume that any position in the simulated mountainous landscape corresponds to a particular

5  habitat containing a pool of adapted species able to move in a limited elevation range up and down their initial and preferred habitat elevation.

We chose to map landscape connectivity and its distribution by measuring the landscape elevational connectivity (LEC) proposed by Bertuzzo et al. (2016). This metric does not account for a specific species but rather focuses on a pool of species assuming a niche width that can be a percentage of the mountain elevation range or fixed. Here, we use the *bioLEC* Python

10  package to measure the LEC index (Salles and Rey, 2019). The LEC is calculated on the evolving landscape obtained from *Badlands* at discrete time intervals and measures how easily species living in other habitats can spread and colonize a given point.

Considering a 2D lattice made of $N$ squared cells, LEC for cell $i$ (LEC$_i$) is given by

$$\text{LEC}_i = \sum_{j=1}^{N} C_{ji} \tag{5}$$

where $C_{ji}$ quantifies the closeness between sites $j$ and $i$ with respect to elevational connectivity. $C_{ji}$ measures the cost for a given species adapted to cell $j$ to spread and colonise cell $i$. This cost is a function of elevation and evaluates how often species adapted to the elevation of cell $j$ have to travel outside their optimal species niche width ($\sigma$) to reach cell $i$ (Fig. 1b) assuming that species niche is represented by a Gaussian function of the elevation.

Estimation of $C_{ji}$ requires computation of all the possible paths $p$ from $j$ to $i$ and is defined as the maximum closeness value along these paths. Following Bertuzzo et al. (2016), $C_{ji}$ is expressed as:

$$-\ln C_{ji} = \frac{1}{2\sigma^2} \min_{p \in \{j \to i\}} \sum_{r=2}^{L} (z_{k_r} - z_j)^2 \tag{6}$$

where $p = [k_1, k_2, ..., k_L]$ (with $k_1 = j$ and $k_L = i$) are the cells comprised in the path $p$ from $j$ to $i$.

Eq. 6 is solved for each cell $j$ using Dijkstra's algorithm (Dijkstra, 1959) with diagonal connectivity between cells. The approach in *bioLEC* is based on scikit-image Dijkstra's algorithm (van der Walt et al., 2014). For each cell $j$, the algorithm builds a Dijkstra tree that branches the given cell with all the cells defining the simulated region. Edge weights are set equal to the square of the difference between the considered vertex elevation ($z_{k_r}$) and $z_j$. The least-cost distance between $j$ and $i$ is then calculated as the minimum sum of edge weights obtained from the cells along the shortest-path (Fig. 1b).

Calculation of LEC values over the entire simulated region ($\sim$1 M points in this study) is slow. Here we use the parallel strategy available in *bioLEC* package where Dijkstra trees for all paths are balanced and distributed over multiple processors using message passing interface (MPI). Using this approach, LEC computation on the synthetic landscapes obtained in this study is less than 4 minutes when distributed over 240 processors.

## 2.3 Experimental setting

Our experimental design consists of a 100×100 km surface at a resolution of 100 m (Fig. 1a). The initial topography is a flat area over which random noise ($\leq$ 1 m) is applied. To estimate landscape connectivity through a complete orogenic cycle, simulations are run for 10 Myr and an uniform uplift of 1 mm/yr is applied during the first 5 Myr (*e.g* mountain building phase) on all surface nodes with the exception of boundary points that remain fixed over the runs (Fig. 1a). A second phase, corresponding to the cessation of tectonic activity over the remaining 5 Myr, simulates the lowering of interfluves and the erosional decay of the mountain system (*e.g* mountain relaxation phase).

In this study and for simplicity, the SPL parameters (Eq. 1) are set constant and do not change between simulations. We assign an uniform erodibility coefficient $\kappa_e$ of 8.e$^{-6}$/yr and exponents $m$ and $n$ are set to 0.5 and 1 respectively; these exponents values are commonly used for eroding fluvial systems (Whipple and Tucker, 1999; Ferrier et al., 2013). For the hillslope processes (Eq. 2), a constant diffusion coefficient of 0.1 m/yr is chosen.

To estimate the influence of climate on geomorphic changes and in turns on landscape connectivity, we perform two simulations (Fig. 1a) with (1) an uniform precipitation rate of 1 m/yr over the simulated 10 Myr and (2) an orographic precipitation model considering a prevailing wind direction from the South with a wind speed of 0.5 m/s and other orographic parameters set within the range proposed by Smith & Barstad (Smith and Barstad, 2004). In both cases, simulated mountain ranges are characteristic of dendritic, erosional landscapes.

Computation of LEC over time requires the definition of species niche width ($\sigma$). In the study conducted here, we limit our analysis to the case where all species have the same niche width at specific time assuming a ratio $\sigma/(z_{max} - z_{min})$ constant. As the landscape changes through time, the elevation range $[z_{min}, z_{max}]$ is modified and so is species niche width. Therefore, we implicitly assume that characteristic response of landscape to tectonic and climatic forces happens on temporal scale much longer than the effective evolution and adaptation of individual species (Mokany et al., 2012; Steinbauer et al., 2016).

## 3 Results

### 3.1 Connectivity distribution under uniform conditions

In this first set of results, we consider a geographic domain, free of environmental gradients, under a constant precipitation rate. As the region is uniformly uplifted at a rate of 1 mm/yr during the first 5 Myr, landscape dissection by riverine and hillslope processes leads to valleys deepening and the development of self-organized and stable large drainage basins (Fig. 2a).

Following the initial orogenic phase of plateau uplift and landscape incision, rivers longitudinal profiles exhibit concave upward shape, a common signature of bedrock river systems (red lines in Fig. 3b). This typical shape is a direct outcome of the detachment-limited stream power law model and is the result of the integrated effect of tectonic, climatic and bedrock erodibility. After 4 Myr of evolution, the landscape has fully adjusted to the imposed constant climatic and tectonic forcing, as shown by the increasing linearity of rivers $\chi$-plots (blue lines in Fig. 3b) (Perron and Royden, 2013). After the cessation of uplift at 5 Myr, erosion induces the rapid lowering of elevations (Fig. 2a) and slope gradients (Fig. 3d) through valleys widening that ultimately leads to a low-relief peneplain.

From an elevational niche perspective, all connectivity (*e.g.*, migratory) paths on a flat landscape are equal. This is not the case on a complex landscape where connectivity can only occur along a network of corridors providing species with their elevational requirements (Smith et al., 2014; Elsen and Tingley, 2015). The mapping of spatial distribution of LEC (Fig. 2a) and associated gradients of LEC (Fig. 3a & c) reveals a migration of maximum LEC regions - during the initial orogenic phase - from highest elevations (*e.g*, uplifted plateau) to mid-elevations as the landscape stabilises.

This redistribution leads to two regions of maximum LEC with the highest one attributed to the symmetrical nature of the simulated synthetic landscape. During the relaxation phase, the LEC peak migrates to lower elevations as the floor of the valleys widens (Fig. 2b). After dissection of the uplifted plateau, the model predicts that higher LEC are found at intermediate to high elevations (Fig. 3a thick blue line). This result agrees with the hump-shaped elevational gradients of mean species richness widely observed in nature (Grytnes and Vetaas, 2002; Colwell et al., 2004; Brehm et al., 2007). We also see that for areas at similar elevation, connectivity values could differ significantly as shown by standard deviation curves (highlighted

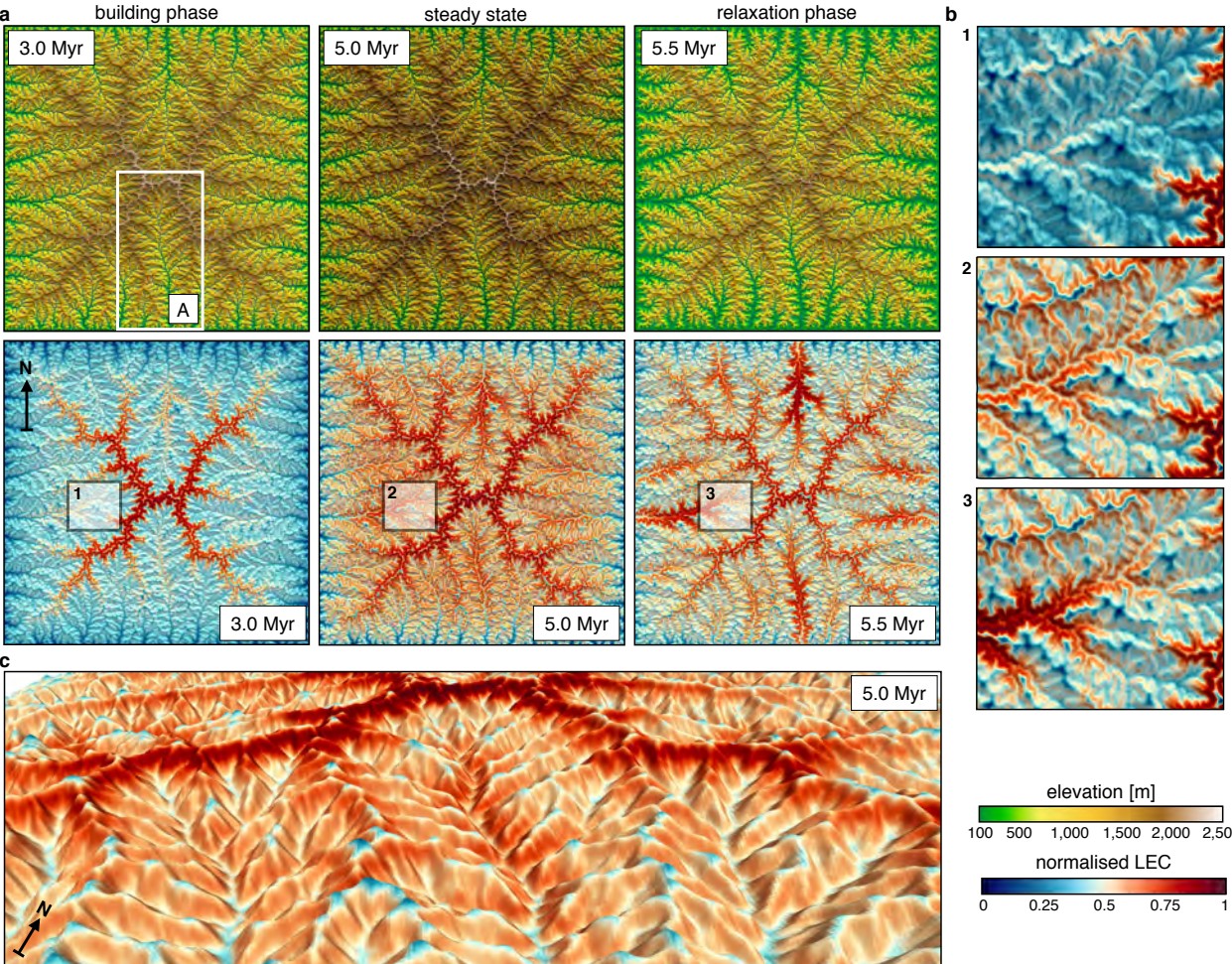

**Figure 2.** Outputs from the uniform precipitation (1 m/yr) model. **a)** Top row presents the evolution of the landscape at 3 time steps during the mountain building phase (induced by an uniform uplift of 1 mm/yr), just before the cessation of uplift at 5.0 Myr (landscape is at steady state, *i.e.* erosion balances uplift) and during the early phase of mountain collapse (5.5 Myr). Bottom row shows the spatial distributions of normalised LEC for the 3 considered time steps. **b)** Detailed maps presenting changes of normalised LEC for regions 1, 2 & 3 defined in a) and illustrating the geomorphic controls on LEC variability over time. **c)** 3D view of spatial patterns of LEC index at 5 Myr with generally low LEC for valleys and minor peaks lower on the ridges and high LEC on ridge flanks.

blue regions at 5.0 and 6.0 Myr in Fig. 3a & c). It suggests that using only elevation as a predictor of landscape connectivity is unlikely a good proxy.

We also observe that, irrespectively of the scale of observation, LEC distribution follows a similar pattern for specific time intervals even if the considered domains traverse different elevational extent. For example, the result in Fig. 2c shows similar mid-elevations maximum LEC (*i.e*, peak of diversity) over both valley, catchment and regional scales.

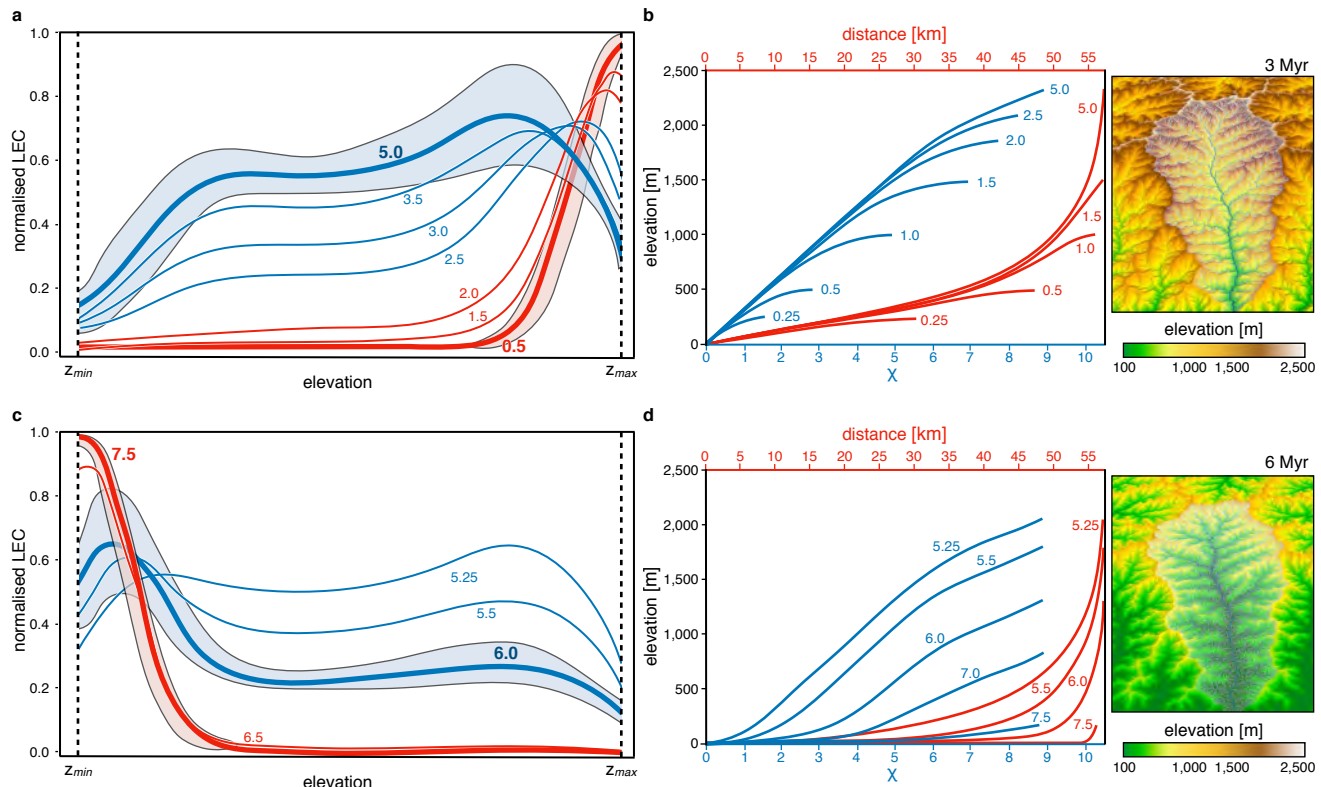

**Figure 3.** Analysis of uniform precipitation results. **a)** & **b)** relate to the mountain building phase (0–5 Myr) and **c)** & **d)** to the relaxation phase (5–10 Myr). Plots in **a)** & **c)** illustrate the elevational gradients of normalised LEC over time (corresponding numbers in Myr for each lines). Blue/Red lines represent average of normalised LEC within elevational bands. Blue/Red areas are the standard deviation computed based on normalised LEC values as a function of site elevation at 5 & 6 Myr (blue areas) and 0.5 & 7.5 Myr (red areas). For each plots **a)** & **c)**, line colours reflect different stages of landscape evolution. In **a)** change from red to blue marks the disappearance of uplifted plateau. In **c)** change from blue to red relates to the erosion of 50% of the maximum elevation obtained at steady state. These 2 plots have a moving horizontal scale as $z_{min}$ and $z_{max}$ change with time. **b)** & **d)** show main stream temporal evolution for catchment area (region A) defined in Fig. 2a and displayed at 3 and 6 Myr respectively (right panels). For the considered catchment, we show the extracted information from main channel profiles based on $\chi$-plot analysis (Perron and Royden, 2013) (blue) and longitudinal river profiles (red) over time (associated time in Myr is given for each lines).

Our results show that under uniform tectonic and climatic forcing, landscape dynamic exerts a first-order control on landscape connectivity distribution which is mostly distributed at mid-elevation and peaks on highest flanks when geomorphic steady state is reached (Fig. 3a thick blue line). The peak position is related to the symmetrical nature of the chosen boundary conditions and species niche width ($\sigma = 0.1$) as shown in section 4.1.

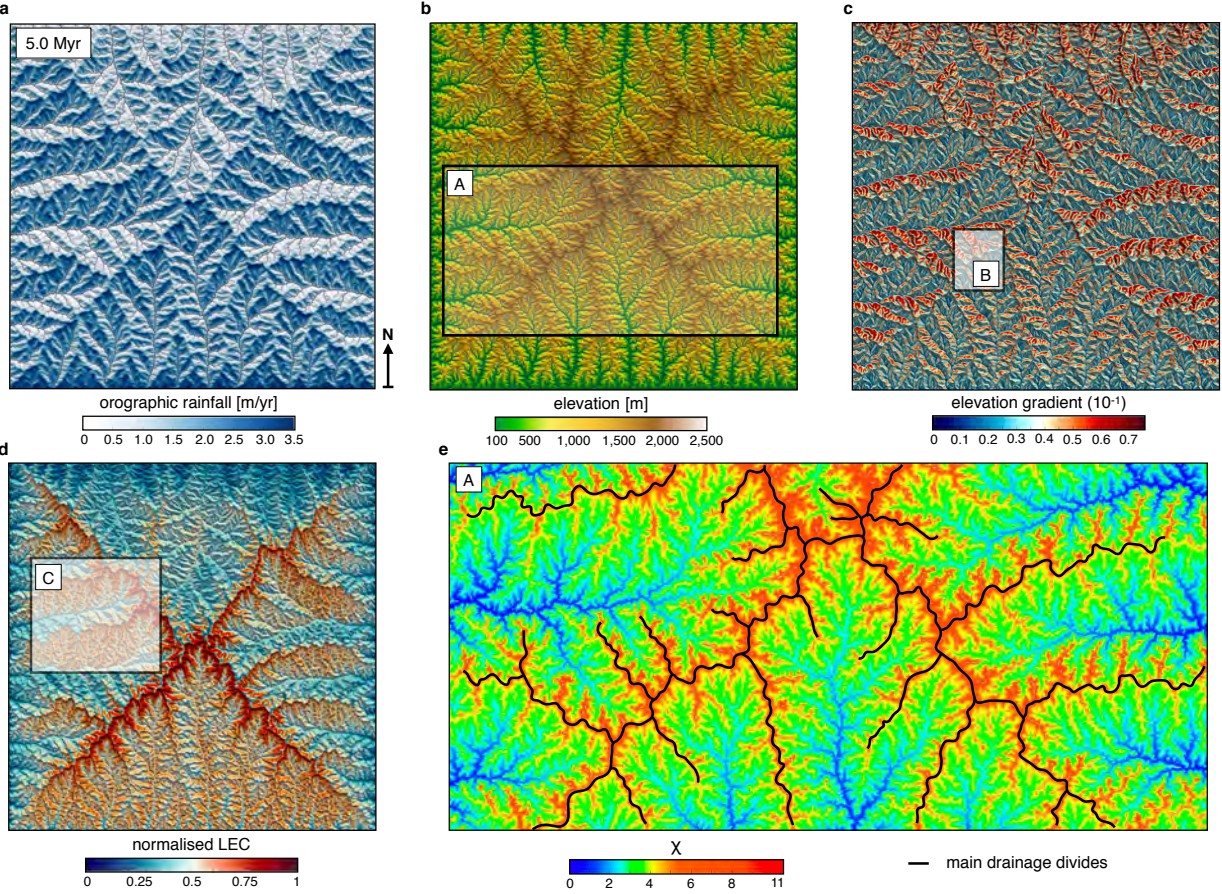

**Figure 4.** Generated outputs of the coupled landscape evolution and orographic precipitation models at the end of the uplift phase (5.0 Myr).
**a)** Induced rainfall map from the orographic model where precipitation is determined by prevailing upslope winds (coming from the South) and by the moisture content of the air. Rainfall distribution is characterised by precipitation maxima on the windward face of the divides. **b)** Resulting elevation grid where the topographic evolution of the mountain ranges is affected by spatial pattern of precipitation and tectonic. During the building phase, main drainage divides continuously migrate towards the drier side and asymmetric topography develops with steeper regions **c)** on the leeward side of the ranges. **d)** Resulting normalised LEC showing specific patterns of distribution with higher values obtained on windward facing catchments. At this timestep, lower LEC are generally found in valleys and mountain tops and higher LEC on the flanks of both the mountain tops and main valley systems. **e)** $\chi$ map (Perron and Royden, 2013; Willett et al., 2014) for inset A (shown in b) highlighting discontinuous $\chi$ across drainage divides, with larger $\chi$ values on the leeward facing catchments (*victim* basins) compared to the windward ones (*aggressor* basins). Insets B and C refer to the region used in the analysis performed in figure 5 and 9 respectively.

## 3.2 Orographic effect on patterns of landscape connectivity

To explore the influence of heterogenous precipitation on connectivity, we design a new set of experiments using a coupled climate and landscape evolution model where precipitation responds through space and time to the build up of topography (*i.e*, orographic precipitation, Fig. 4a & b).

In this experiment, a prevailing northward directed wind condition creates specific distribution of rainfall patterns across the mountain range with precipitation maxima occurring predominantly on windward facing areas. Induced changes in hydrological regime drive drainage divides migration (Fig. 5a) and the development of asymmetric topography (Roe et al., 2003; Anders et al., 2008). Topography on the leeward side (*rain shadow* area) is formed by tectonic uplift with limited erosion, whereas topography on the windward side is predominantly erosional. Over geological time-scales, differential distribution of rainfall

influences overall topographic evolution of the simulated mountain range. First, it affects river profiles by changing concavity (Roe et al., 2003) and reducing erosional rates on the drier regions. As a consequence, a decrease in discharge produces steeper slopes to compensate the imposed uniform uplift rate (Fig. 4c). Orographic rain also implies continuous disequilibrium conditions (Willett et al., 2014; Whipple et al., 2016) affecting river networks through reorganization and concomitantly changing the topologies and geometries of drainage basins (Fig. 4e).

Similar to the uniform precipitation model, we observe a temporal migration of LEC maximum from the uplifted plateau towards the bottom of the valleys. However, we find some specific features driven by the geomorphic responses to orographic precipitation (as shown by the LEC distribution at steady state between Fig. 2a and Fig. 4d) with a clear correlation between sites with high/low connectivity areas and windward/leeward facing catchments. It shows how uplift and its effect on regional climate influence landscapes connectivity with development of geographic connections or barriers as topography changes.

The study of two specific catchments sharing a common drainage divide highlights further these effects. In Figure 5, we note that both leeward and windward catchments exhibit similar hump-shaped patterns of LEC, indicative of mid-elevation peaks, with higher connectivity on the windward facing area compared to the leeward facing one. Temporal evolution analysis through the complete orogenic cycle shows optimum landscape connectivity during the mountain building phase at mid- to high-elevations as illustrated by the broader ranges of LEC values at 2.5 Myr compared to the ones at 5.5 Myr (Fig. 5b).

## 4 Discussion

The results described above quantify the temporal and spatial evolution of landscape connectivity patterns during a complete orogenic cycle considering both uniform and orographic precipitation scenarios. In this discussion section, we build upon the work from Bertuzzo et al. (2015, 2016). First, we test the influence of different boundary and species niche width conditions on landscape connectivity. Then, we explore the relationship between landscape elevational connectivity and biodiversity using a

similar metacommunity model as the one proposed in Bertuzzo et al. (2016) and discuss the possible roles of geomorphological changes on patterns of species richness. By defining isolated regions as places with low connectivity, we then analyse how they evolve during simulated orogenic cycles.

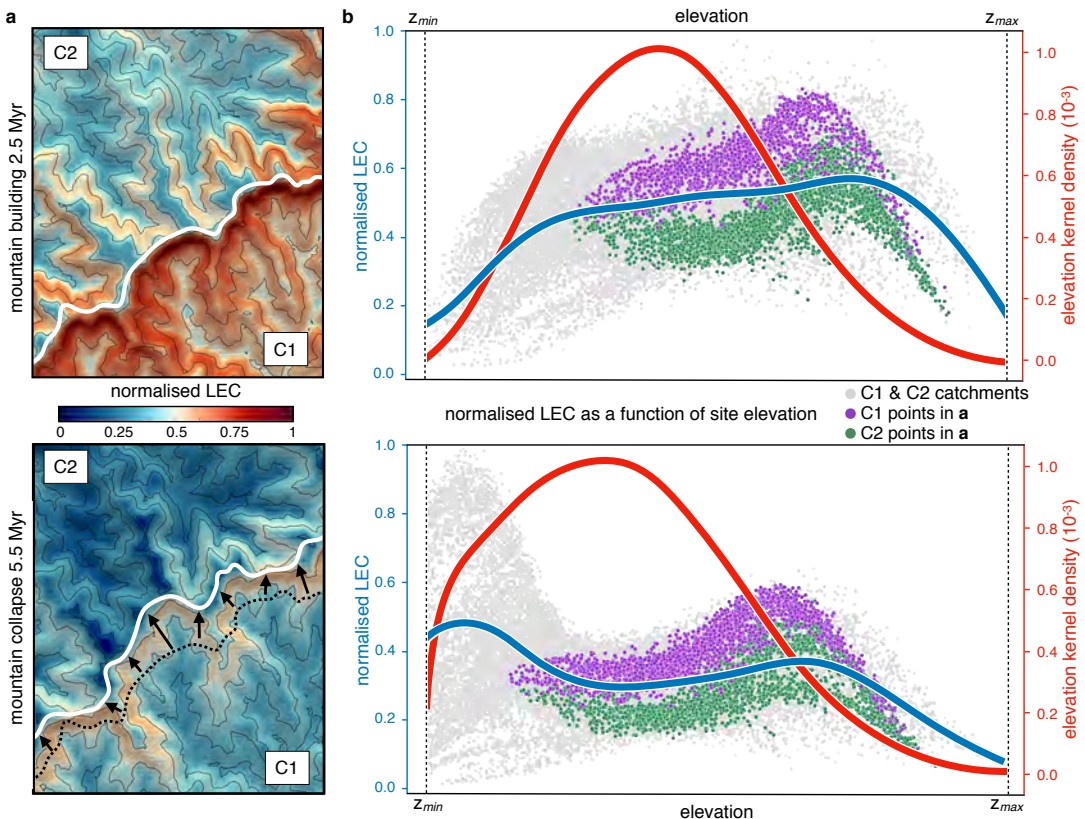

**Figure 5.** Temporal evolution of normalised LEC distribution patterns between windward (C1) and leeward (C2) catchments sharing a common divide (region B defined in Fig. 4c). **a)** Normalised connectivity maps at 2.5 Myr and 5.5 Myr respectively, white line represents the drainage divide position at current time step. Black dot line in bottom panel shows divide position at 2.5 My and black arrows its migration direction. Contour lines are defined every 250 m and range between 500 and 1,750 m. **b)** Plots of normalised LEC (gray dots), average line of LEC within elevational bands (blue lines) and elevation kernel density (red lines) for the entire simulated domain at chosen time intervals. Normalised LEC for points belonging to the windward and leeward catchments (C1 and C2 respectively) are highlighted. The normalised LEC exhibits a typical pattern with higher values and broader ranges during mountain building phase at mid to high elevations.

## 4.1 Impact of forcing conditions and species niche width on connectivity mapping

In our experimental setting, we have deliberately chosen a simple model of an evolving landscape similar to existing analogue and numerical models (Bonnet and Crave, 2003; Bonnet, 2009; Willett et al., 2014; Salles and Hardiman, 2016). Under such conditions, the resulting landscape is symmetric with the formation of a mountain range composed of four low angled ridges ending at the four corners of the model. In this section, we modify slightly these initial tectonic boundary conditions and perform a new simulation with outlets along only one of the domain boundary (Castelltort and Yamato, 2013). The other landscape parameters (erodibility value $\kappa_e$, SPL exponents $m$ and $n$ and hillslope diffusion $\kappa_d$ defined in section 2.3) remain unchanged and the simulated landscape is again purely erosional (similar to the homogeneous model proposed in Roy et al.

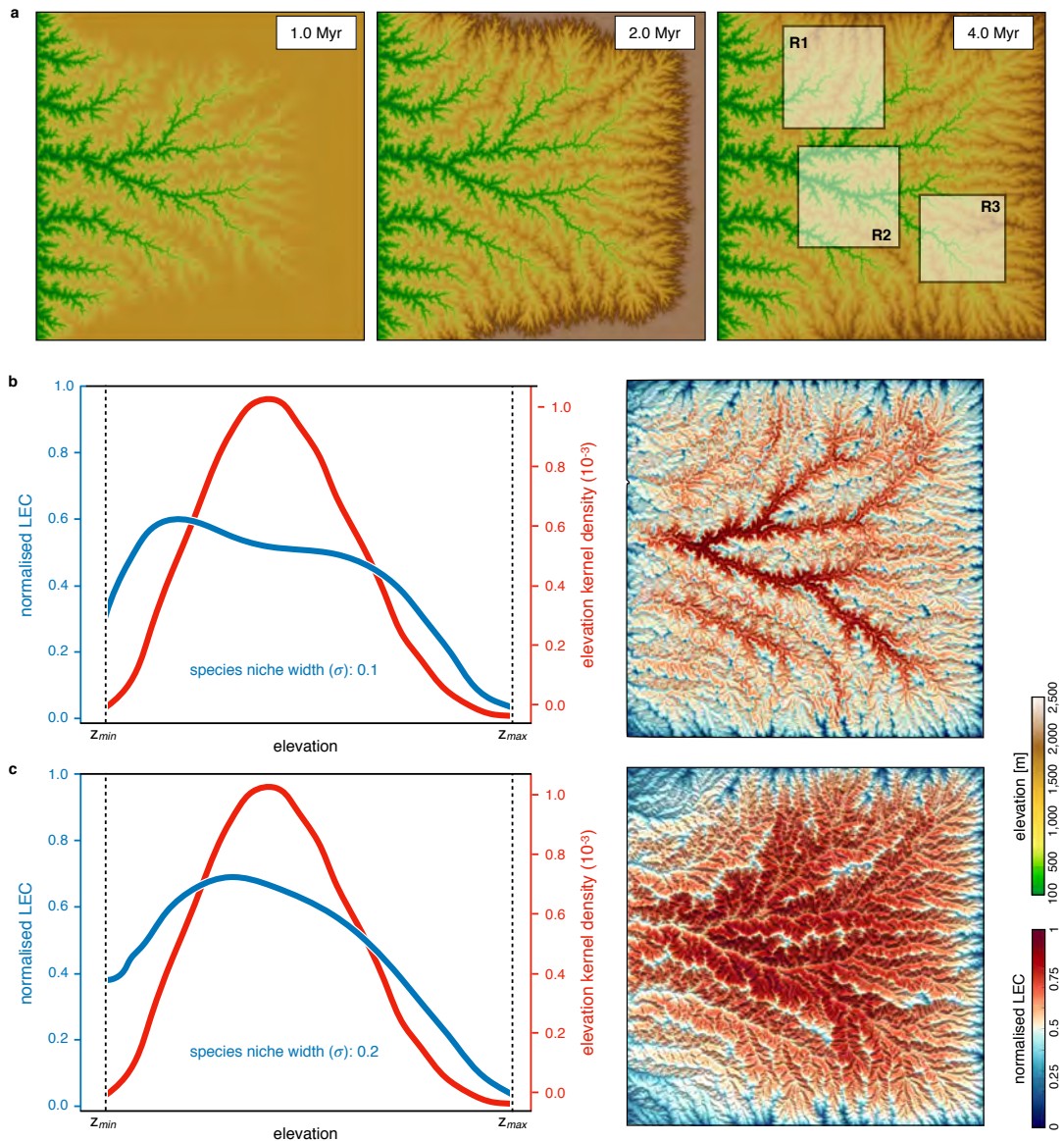

**Figure 6.** Outputs from the modified uniform precipitation model with only one fixed boundary along the western border of the simulated domain. **a)** Maps of the evolution of the landscape at 3 time steps during the growth phase (induced by an uniform uplift of 1 mm/yr). Panels **b)** and **c)** show the results obtained with two ratio for $\sigma/(z_{max} - z_{min})$ of 0.1 and 0.2 respectively. In both cases, we present on the left the average line of LEC within elevational bands (blue lines) and elevation kernel density (red lines) for the entire simulated domain at steady-state (5 Myr) and on the right, the corresponding normalised connectivity map.

(2015)). To evaluate how LEC will change for constructional landscapes (Craw et al., 2015, 2019) requires different tectonic conditions and is beyond the scope of this paper.

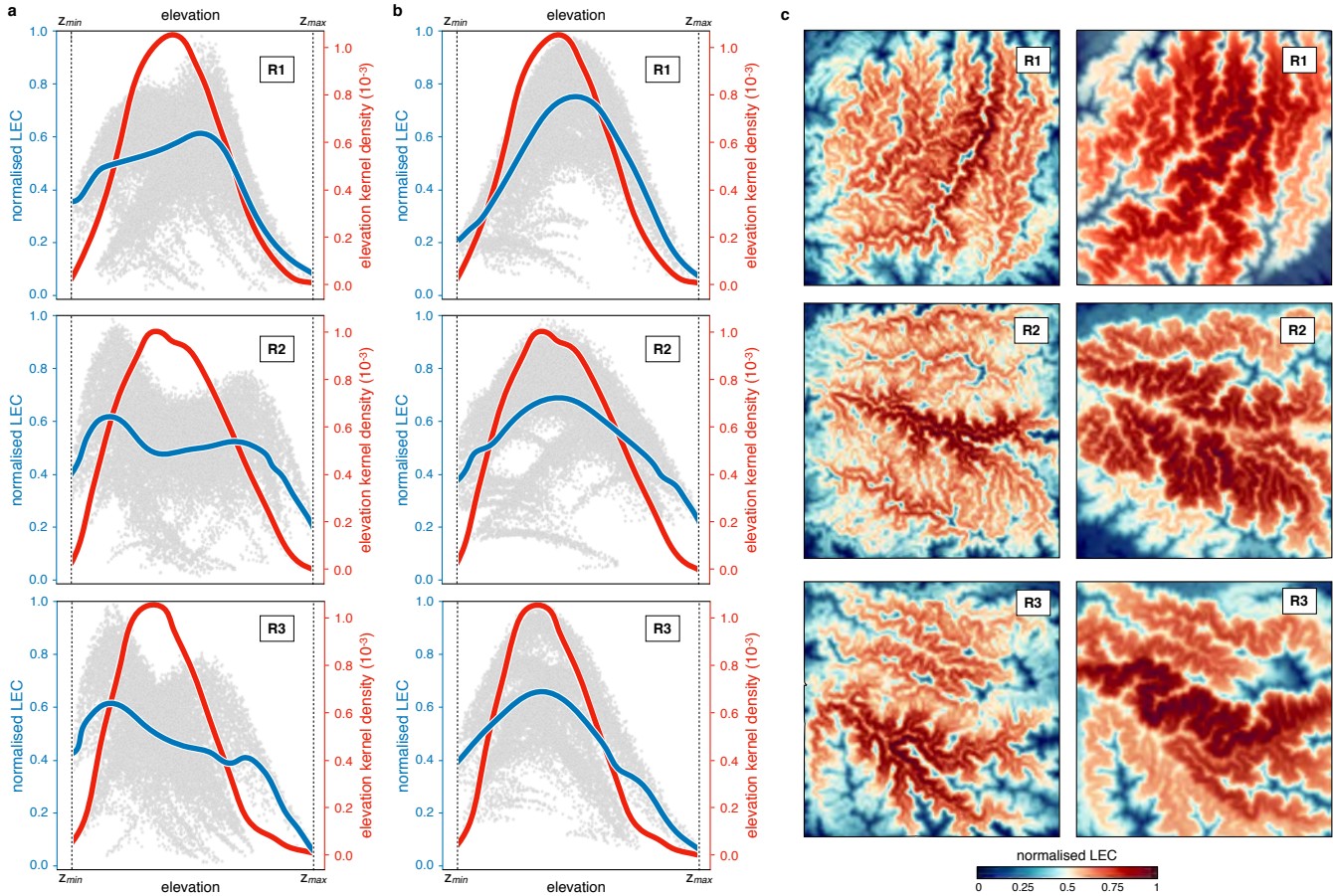

**Figure 7.** Connectivity results for the regions (R1, R2 and R3) presented in Fig. 6a. Panels **a)** and **b)** show corresponding average line of LEC within elevational bands (blue lines) and elevation kernel density (red lines) for two species niche width ratio of 0.1 and 0.2. Panel **c)** presents the normalised connectivity map for the two width ratios and for the three regions.

The general pattern of geomorphological evolution under uniform precipitation is presented in Fig. 6a for the first 5 Myr and it corresponds to the mountain building phase. As for the previous cases, the evolution of the experiment involves a growth phase followed by a steady-state phase (Bonnet and Crave, 2003; Castelltort and Yamato, 2013). During the initial plateau uplift (*e.g.*, growth phase), some topographic incisions form along the western open border of the model. As uplift continues, these incisions grow and propagate inward until there is complete dissection of the plateau (third map on the right hand side of Fig. 6a). The obtained fluvial landscapes display dendritic networks and rivers longitudinal profiles present a concave upward shape characteristic of bedrock river systems (as for Fig. 3b).

To evaluate the impact of these new boundary conditions on connectivity calculation, we compute the LEC at steady state (Fig. 6b) over the entire domain and for three quadrants (Fig. 7). In addition, we test two values of 0.1 and 0.2 respectively for the species niche width ratio ($\sigma/(z_{max} - z_{min})$) defined in Eq. 6. As shown in the results (section 3) and for the Swiss

Alps in Bertuzzo et al. (2016), the frequency distributions of the elevation is hump-shaped regardless of the spatial extent of the considered domain (red lines representing the entire domain in Fig. 6 or three smaller regions in Fig. 7). From the results presented in Fig. 7, we find that connectivity distribution varies significantly with respect to variations of niche width $\sigma$. As the niche becomes wider (higher $\sigma$), the connectivity maps (Fig. 6c and 7c) reveal a clear spatial pattern with valleys and mountain

tops characterized by lower LEC (Bertuzzo et al., 2016). As $\sigma$ increases, any point on the map is less constrained by elevational barriers, and thus LEC is mostly controlled by the abundance of sites with similar elevation in the simulated domain (*i.e.*, the elevation frequency distribution as shown by the blue lines in Fig. 6c and 7b). Bertuzzo et al. (2016) show that for very large $\sigma$, LEC becomes insensitive to elevation and elevational gradients tend to flatten out.

## 4.2 LEC as a measure of biodiversity

Bertuzzo et al. (2016) have shown that when applied to real landscapes, the LEC metric predicts well the $\alpha$-diversity simulated by full metacommunity model. Here we perform a similar analysis on our simulated landscape and we estimate for both scenarios (uniform and orographic rain) the correlations between LEC and simulated local species richness ($\alpha$ diversity).

To do so, we use a zero-sum metacommunity model (Hubbell, 2001) where local communities ($N$) are defined on an regular 2D elevation mesh. The zero-sum assumption states that each local community is saturated at all time, which means that the

total number of individuals $n$ in a particular point is constant over time. In such case and for any given time, the entire region is made of a population of $N \cdot n$ individuals where $N$ is equivalent to the number of points on the grid. The population number is made of different species which have different elevational niches. Each of these niches describes the competitive ability of particular species with elevation and are defined with a Gaussian function following Rybicki and Hanski (2013) (Bertuzzo et al., 2016):

$$\mathrm{f}_i(z) = \mathrm{f}_{\mathrm{max}_i} \mathrm{e}^{-\frac{(z - z_{\mathrm{opt}_i})^2}{2\sigma_i^2}} \tag{7}$$

with $\mathrm{f}_i(z)$ the competitive ability of species $i$ at elevation $z$. The elevation $z_{\mathrm{opt}_i}$ is the optimal elevation for which the competitive ability equals $\mathrm{f}_{\mathrm{max}_i}$ and $\sigma_i$ is the niche width for species $i$ as defined in Eq. 6. In addition to the zero-sum assumption, we suppose that all species have the same niche width $\sigma_i = \sigma$ and maximum competitive ability $\mathrm{f}_{\mathrm{max}_i} = \mathrm{f}_{\mathrm{max}}$ as well as similar rates of dispersal, death and fertility.

The ecological interactions between individuals follow the same approach as the one proposed in Bertuzzo et al. (2016). For any given time step and on each local community, a randomly chosen individual dies. Based on the zero-sum assumption, this dead individual needs to be replaced and two strategies are possible. First, the offspring can come from the pool of individuals living in the vicinity of the local community (*i.e*, coming from the community itself or from one of the neighbouring ones). In this case, the chosen offspring is selected with a probability proportional to the values of $\mathrm{f}_i(z)$ for all the individuals living in

the vicinity of the local community (where $z$ is the elevation of the considered local community). Second, the offspring is an individual belonging to a new species that does not already exist in the system. This second option is probabilistically selected at every time step and allows us to model both speciation and immigration from external communities (Hubbell, 2001; Chave

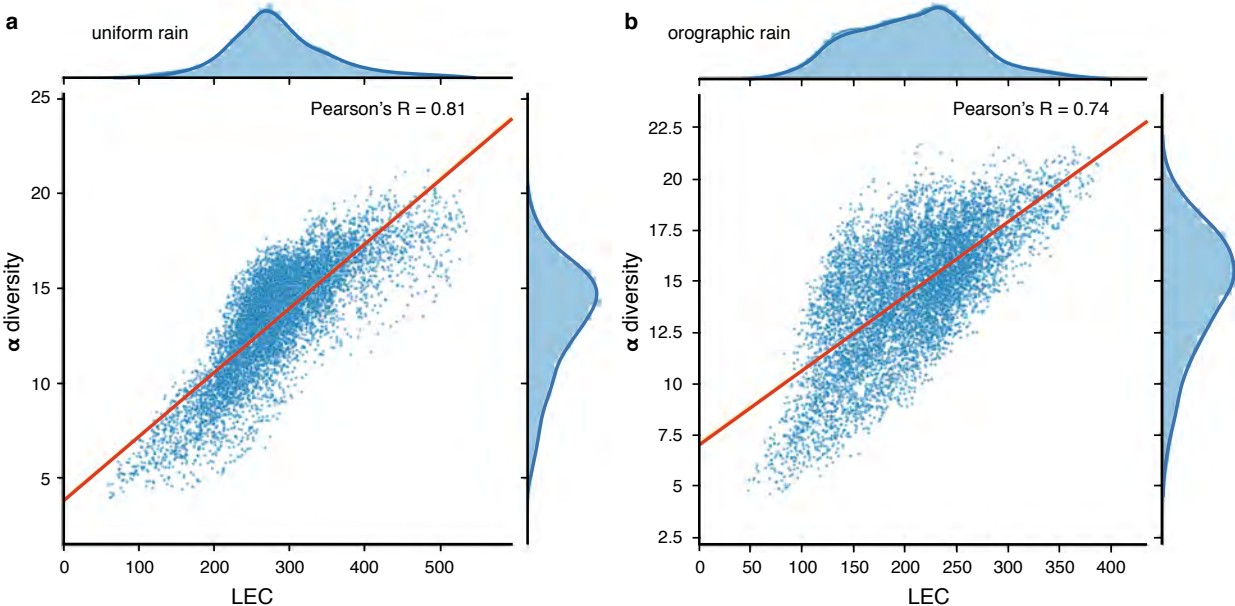

**Figure 8.** LEC versus local species richness ($\alpha$ diversity) for simulated landscape at steady state (5 Myr) under uniform rainfall **a)** and orographic precipitation **b)**. The $\alpha$ diversity patterns result from the zero-sum metacommunity model described below. Correlations between $\alpha$ diversity and LEC are presented (blue dots) as well as regression and kernel density fits for each case. We also provide the resulting Pearson's coefficients for correlation between $\alpha$ diversity and LEC values.

et al., 2002). When a competitor from a new species enters the system its optimal elevation $z_{\mathrm{opt}_i}$ is drawn from a uniform distribution spanning twice the relief of the system to avoid edge effects (Bertuzzo et al., 2016).

This metacommunity model is applied on the elevation grids obtained from our landscape evolution model. In all simulations, the system is initially populated by one single species and is ran until a statistically steady state is reached ($\sim 10^4$ generations, where a generation is $N \cdot n$ time steps). As in Bertuzzo et al. (2016), we consider periodic boundaries conditions. The parameter $f_{\mathrm{max}}$ does not affect the system dynamics and is, without loss of generality, set to 1.

Fig. 8 presents the relationships between calculated LEC and local species richness for the simulated landscapes at steady state. Overall we have Pearson's correlations (R) >70% which is considered to be relatively strong. These results are in agreement with Bertuzzo et al. (2016) study and consolidate the idea that the LEC calculation is able to predict well the $\alpha$ diversity especially in the case of the uniform rainfall. Therefore if LEC captures the regional variability of communities diversity one can deduce from its distribution the temporal and spatial evolution of species richness.

We provide in Fig. 9 a statistical summary of the LEC evolution for the two climatic scenarios. LEC predicts biodiversity peaks when the landscape reaches a geomorphological steady state, and maximum species richness is observed at mid-elevation (Fig. 9a – $0.7 < z/z_{max} < 0.9$) as observed in nature (Grytnes and Vetaas, 2002; Colwell et al., 2004; Brehm et al., 2007). Interestingly, when using a coupled climate-landscape model, windward facing regions host higher biodiversities. During

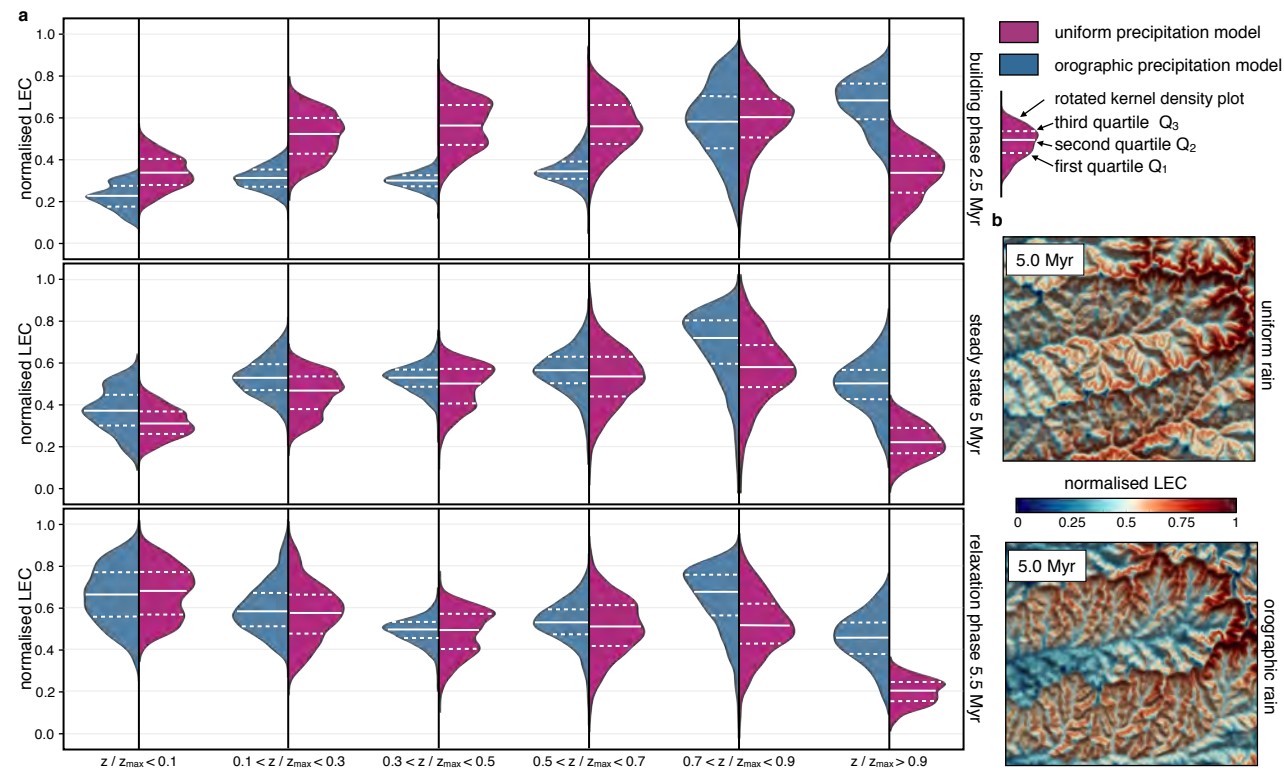

**Figure 9.** Temporal and spatial statistical analysis of normalised LEC based on categorical elevation data (defined within six bands of $z/z_{max}$) for both the uniform and orographic precipitation models. **a)** Violin plots depicting kernel density estimation of the normalised LEC distribution. White dashed lines in the violin plots show the location of the lower quartile ($Q_1$) and the upper quartile ($Q_3$). Solid white lines define the median ($Q_2$) of the data set. **b)** Comparison of spatial distribution of normalised LEC at 5 Myr for the two considered climatic scenarios. The data used for the statistical analysis presented in a) have been extracted from these maps. The considered area corresponds to the highlighted region C defined in Fig. 4d.

relaxation phase, when tectonic forcing ceases, peak in species richness shifts from mid to low-elevations (bottom graph in Fig. 9a). From the figure, we infer that mapped areas showing strong connectivity values (*i.e.*, high LEC), within a given elevational band, have a direct effect on the biodiversity they host (Rahbek, 1997; McCain, 2007). We also hypothesise that these areas represent regions of high local species richness as local communities can be gathered from a more diverse regional
5 pool of species that are fit to live at a similar elevation (Romdal and Grytnes, 2007).

In summary, LEC calculation can be used to estimate in a simple way the temporal and spatial evolution of biodiversity distribution induced by changes in landscape structure over geological time. Based on LEC distribution at steady state, we found that species richness in mountainous landscapes reaches its peak at mid-elevation not only because this is where the maximum surface area available to species is located, but also because species can move up or down to accommodate climate
10 warming or cooling respectively.

There are many biologic and abiotic properties that drives species richness in mountainous landscapes (Hoorn et al., 2010, 2013; Giezendanner et al., 2019). For example, mountains provide species with a rich variety of environmental conditions over restricted surface areas (*e.g.* range of temperature, solar irradiation, wind exposure, moisture and rainfall, soils thickness and composition to cite a few). In this study, we have limited our exploration to the role of morphological changes imposed by uniform tectonic, riverine and climatic processes and we found that these changes set alone could explain to the first order the biodiversity found in mountainous regions.

### 4.3 Dynamic of geomorphic-driven isolation

From the LEC values, one can derived geomorphic-driven isolation by finding spatial regions of low LEC compared to their surrounding areas. Here, we apply an approach similar to a depression-filling algorithm (Barnes and Lehman, 2013) using the LEC values instead of the elevation. Isolated areas in that sense, represent depressions or inwardly-draining regions of the LEC map which have no outlet. To prevent coalescence of these isolated regions (*i.e*, small depressions within bigger ones), one can apply a threshold on the filling limit of the LEC map. Increasing such threshold produces a smaller number of isolated regions but with larger areas.

In Fig. 10, we evaluate through time these areas of low LEC values (normalised LEC below 20%) and analyse series of metrics: mean LEC, mean area and mean elevation. Our results show that the competition between mountain-building and erosion increases the number of isolated regions during the orogenic phase (Fig. 10a). Orographic precipitation fosters faster isolation than the uniform precipitation model (red lines trends in Fig. 10a) especially on the steeper and drier leeward sides (*rain shadow* zones) of the drainage divides (Fig. 10b). For the uniform rain model, we observe a decrease in the average area of these isolated regions concomitant with catchments stabilisation and emergence of the steady state (light grey line in Fig. 10a top graph). For both precipitation scenarios, the cessation of tectonic activity at 5 Myr results in a rapid decrease in the number of isolated regions associated with an initial episode of augmentation of mean area that last no more than 250,000 years. We attribute this peculiar effect to the fast widening of valleys high- and mid-elevation zones related to the adjustment of catchments morphology and induced by the advective slope retreat predominant across the valley heads (Pelletier, 2011). Hence, our model predicts an increase in isolation during the mountain building phase, and a decrease during the relaxation phase.

Based on the results from section 4.2 and as high LEC regions represent – to the first order – places of higher biodiversity, we could take the argument further and propose that low LEC areas should potentially correspond to places where species get trapped into isolated refuges and such ecological niches should favour speciation and endemism.

Not surprisingly, it has been found that the potential for isolation increases in more topographically diverse areas (Ali and Aitchison, 2014; Gillespie and Roderick, 2014). In recent years, several studies have shown that isolation induced by uplift and climatic processes plays a central part in shaping biodiversity (Wang et al., 2009; Craw et al., 2015) and mapping of geomorphic-driven low landscape connectivity could potentially highlight these areas of speciation hotspots. Therefore the LEC calculation could provide an efficient way to identify and manage regions of higher conservation value.

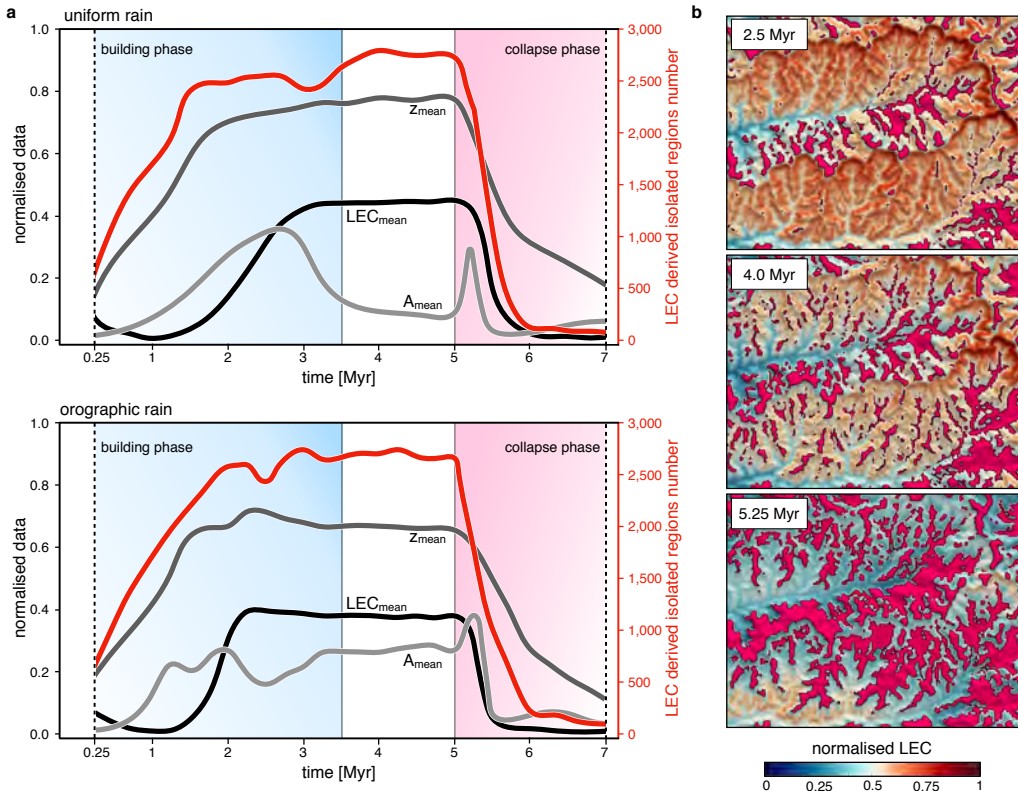

**Figure 10.** Temporal evolution of the isolating effect of mountain geomorphology derived from LEC computation. The method consists of mapping and integrating over space and time regions of lower connectivities (normalised LEC < 20%) surrounded by higher ones. **a)** Analysis of isolated regions for the uniform and orographic precipitation simulations (top and bottom panels respectively). Each panel presents the mean elevation ($z_{mean}$ – dark grey), LEC ($LEC_{mean}$ – black) and area ($A_{mean}$ – light grey) for the isolated regions as well as the total number of extracted regions (red line) through time. **b)** Normalised connectivity maps showing for the orographic model the distribution of isolated regions (magenta areas) at 3 time intervals. The considered maps correspond to the highlighted region C defined in Fig. 4d. Both simulations exhibit a similar trend with increase in number of isolated regions during the building phase (shaded blue area) followed by a period of stabilisation and a rapid decrease during the relaxation phase (shaded red area). We note a sharp increase in mean area at 5 Myr attributed to the rapid adjustment of upper valleys slopes consecutive to the cessation of uplift.

Studies have predicted higher endemism at higher elevation as a consequence of increasing isolation with elevation (Steinbauer et al., 2016, 2018). We see similar trend with high-elevation zones less connected from each other and statistically smaller than lower-elevation ones. However, our results also suggest that low connectivity regions form fragmented patches that are distributed across the entire elevational range during an orogenic cycle ($z_{mean}$ in Fig. 10a). Several other parameters and mechanisms not accounted for in this study will promote spatial variations in speciation rates including temperature or biotic interactions (Emerson and Kolm, 2005; Brown, 2014). Nevertheless, it shows that when considering the morphological complexity of mountainous landscapes over geological time, zones of low connectivity are not distributed continuously over

elevation gradients and the associated isolated regions will likely be found at different elevations. Therefore the prediction of increasing endemism with higher elevation might not always be the rule.

# 5 Conclusions

This paper presents a methodology to quantify landscape connectivity under tectonic and climatic forcing. Over geological time scales (million years), tectonics and surface processes conspire to transform flat regions into complex dissected mountainous landscapes. We used the landscape elevational connectivity (LEC) to measure the connectivity between sites of similar elevations (Bertuzzo et al., 2016). We show that this abiotic parameter is able to account for up to 80% of the $\alpha$ diversity predicted by a zero-sum metacommunity model (Hubbell, 2001; Rybicki and Hanski, 2013) and therefore it could potentially explain the first order distribution of biodiversity found in mountainous regions (Ali and Aitchison, 2014; Steinbauer et al., 2018). In the future, we can easily improve the predictive capacity of this metric by integrating fitness parameters other than elevation.

From the LEC calculation, one can quantify the role of geomorphology on landscape connectivity and topography-driven isolation. As the periodicity of isolation and connection dictates evolutionary outcomes, understanding of this dynamic might be used to test models of biological diversification, and to understand species distribution and biodiversity patterns through time. Our results suggest that peaks in species richness for mountainous landscapes can be derived from the analysis of the dynamic organisation of geomorphic features as they evolved in response to tectonic, climatic and erosion processes. These features might be inferred from physiographic metrics such as $\chi$ maps, in-situ observations, river longitudinal profiles or landscape evolution modelling. We also found that geomorphic-driven isolation has the potential to increase rates of speciation over the entire elevational range. This is an important outcome that could potentially lead to reassessment of the spatial viability of existing conservation sites and could help us improve future conservation planning, policy and practice (Mokany et al., 2012).

*Code and data availability.* The results and data presented and discussed in this paper were simulated using *Badlands* model and the connectivity maps were obtained from *bioLEC* Python package. All the model outputs were visualised using Python Matplotlib library for standard graphics, Seaborn data visualization library for statistical graphics and the Paraview software (V 5.2.0) from Kitware, Sandia National Labs and CSimSoft for 3D graphics and 2D maps.

*Author contributions.* T.S., development of the code, design of the experiment, output analysis, and manuscript writing; P.R., output analysis and interpretation, manuscript writing; E.B., development of the code, output analysis and interpretation, manuscript writing.

*Competing interests.* The authors declare no competing interest

*Acknowledgements.* The authors acknowledge the Sydney Informatics Hub and the University of Sydney's high performance computing cluster Artemis for providing the high performance computing resources that have contributed to the research results reported within this paper. We thank Phaedra Upton, the anonymous reviewer and the journal editor for their comments that greatly improved the manuscript.

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
