# Peer review of "Mapping landscape connectivity as a driver of species richness under tectonic and climatic forcing"

_Earth Surface Dynamics, 2019_

## Referee Comment (RC1) · Phaedra Upton (Referee) · 12 Jul 2019

Review of Salles et al. "Mapping landscape connectivity under tectonic and climatic forcing" Reviewer: Phaedra Upton

I was really interested to read this paper which aims to bring together landscape evolution models and speciation via landscape connectivity. Despite some concerns outlined below, this is a very worthy contribution. After revision, this paper will be a well cited link between biology and geomorphology. I can only select minor or major revisions but consider my suggestions to be moderate rather than major revisions.

Specific comments:

I don't think the boundary conditions chosen for the landscape model do justice to the
study. The model used is a block of rock uplifting at a fixed rate with potential outlets on all sides. The resulting landscape is predictably a mountain with four low angled ridges ending at the four corners of the model. It best reflects an island rising out of the sea. The paper that proposed geomorphic controls on elevational gradients of species richness (Bertuzzo et al. 2016) uses a region of the Swiss Alps. With slightly different boundary conditions, for example, outlets along only one boundary (e.g., Roy et al. 2015 JGR ES 120 homogeneous model), I think the model would have produced results that matched the alpine example more closely. The minimum I would like to see is a discussion of why these particular boundary conditions were chosen and how the authors think that the boundary conditions impact on the model.

I'm concerned about some of the geomorphic statements in the paper. This model is completely erosional, yet it is stated that the topography on the leeward side is constructional (page 9). Constructional topography is formed by block movement, for example, a range is uplifted along a thrust fault and a river may then be constrained to flow along the valley formed by that block uplift. An example is Central Otago, New Zealand where rivers flow parallel to the main ranges. The relief associated with the topography in this region is a result of the block movements along the faults. In this model, the whole block is being uplifted as one and any relief that arises is because of differential erosion. The asymmetric topography and reduced erosion on the leeward side in the orographic model is still an erosional topography.

I think the differences between the two models presented here are overstated. Page 10, line 2 "...has profound effects on the geomorphic expressions of the simulated landscape". The two landscapes are different in detail, but they are both dendritic, erosional landscapes.

These last two points raise an interesting question, which may be beyond the scope of this study, is there a significant difference in the LEC for erosional landscapes (e.g., this study, Bertuzzo et al 2016) and constructional landscapes (e.g., Central Otago – Craw et al. 2016, Nat Geosci, 9; or landscapes further north in New Zealand – Craw

et al., 2019, Geomorphology, 336)? Inclusion in the discussion that this model only addresses one type of landscape (erosional) and how other landscapes might differ would add weight to the manuscript.

I'd like to see more like-for-like plots. Figures 2 and 4 are just different enough that it is hard to see the differences between the uniform rain and the orographic rain models. Similarly, an equivalent to figure 5 for the uniform rain model might be useful.

Figure 2: The caption to (c) states "..generally low LEC for valleys and mountain tops and high LEC on the most elevated flanks of the mountain ranges." I don't think this reflects what is being shown. I see low LEC for valleys and minor peaks lower on the ridges, some higher LEC on ridge flanks but the highest LEC along the highest elevations and along river valleys at 5.5Ma. This is quite different to what is shown in Bertuzzo et al (2016) where the highest elevations are very low LEC. I suspect a plot of LEC vs elevation for this model would not be a bell curve centred on mid-elevations but would be dominated by higher elevations – as is my reading of figure 3A. I think this is a function of the landscape model chosen, as outlined above in the first comment. This may also be being accentuated by the colour map chosen. I find figure 2C hard to reconcile with the text, figure 3 and Bertuzzo et al. (2016). Going from red to blue with a small band of white between them means a lot of the grid appears to be very high LEC when maybe it isn't. I would recommend a different colour map be used.

Technical corrections:

Page 1, Line 17: replace 'This competition converts...' with "These competing processes convert...

Line 20: timescales

Line 24: distances

Page 2, Line 3: change geological to tectonic

Line 8: reword 'highest frequencies for mid-elevations'

Line 8: over geological time? Implies a very long time. Reword to entire orogenic system

Lines 8-12: expand to make clearer

Line 19: as outlined above, I don't think the geomorphic processes are changing, there is a difference in rates due to the change in the distribution of the precipitation driving the model, but it is still based on a stream power/hillslope diffusion model, no change in process.

Line 31: allows us to

Page 4, 1st line of caption: check tense here, are run or were ran

Page 5, Line 16: spelling - cycle

Line 17: check tense here, are run or were ran

Line 25: is this a series of simulations or two simulations?

Lines 32-33: references for this assumption

Page 7, lines 20-21: '. . . model predicts that, on average, higher LEC are found at intermediate elevations.' None of the figures really show this, Figure 3a at 5Ma has a broad peak across most of the elevation range and (c) has a peak at very low elevations.

Line 27: is Fig. 2c the correct figure reference?

Lines 29-30: again I don't see an obvious peak at mid-elevations, rather a broad peak across most elevations.

Page 10, 4th line of caption: defined not defines

Page 12, line 3: . . .the offspring can come from the. . .

Line 8: ?allows us to model

Line 9: . . . a uniform

Page 13, line 9: why is this? Because these slopes aren't as steep therefore allowing for more distance with change in elevation?

Page 15, 1st line of figure caption: replace in at end of line with of

Figure 3: Plots (a) and (c) appear to have a moving horizontal scale, that is zmin and zmax change with time. State in the caption for clarity.

Figure 4: (d) is a map of chi over the whole region. Willett et al. (2014) show chi calculated along streams not the whole region. I think it would be much easier to see what the authors are trying to show here if this plot was similar to Willett et al Figure 3 or 4. The scale bar being pushed toward lower values also makes it difficult.

Figure 5b 5.5My. Given that the area shown in (a) is very small compared to the entire domain, I'm surprised by the distribution of the grey dots beneath those of catchment C1 and C2. Do these represent the whole domain? Shouldn't there be many more grey dots around the elevation range of C1 and C2? This plot suggests that the chosen region is unusual compared to the rest of the domain, but this is not evident in figure 5b.

Figure 8: How is an isolated region defined? The text only specifies low LEC and the figure caption lower connectivities. The final two sentences of the caption appear to refer to (a) not (b). The third plot of 8b doesn't look as asymmetric as the second plot of figure 7b.

There are some inconsistencies between the figures.

The region of 4d shown in figures 7 and 8 don't appear to be the same. The third plot of 8b doesn't look as asymmetric as the second plot of figure 7b.

Scale bars for LEC, figure 8 has more white around the median value of 0.5 than in other figures. This makes comparisons difficult.

---

## Short Comment (SC1) · 18 Jul 2019

First, we would like to thank the reviewer, Phaedra Upton, for her comments. Below we provide a point by point response to these comments. We have also attached the updated manuscript.

Specific comments:

SC1: I don't think the boundary conditions chosen for the landscape model do justice to the study. The model used is a block of rock uplifting at a fixed rate with potential outlets on all sides. The resulting landscape is predictably a mountain with four low angled ridges ending at the four corners of the model. It best reflects an island rising out of the sea. The paper that proposed geomorphic controls on elevational gradients

of species richness (Bertuzzo et al. 2016) uses a region of the Swiss Alps. With slightly different boundary conditions, for example, outlets along only one boundary (e.g., Roy et al. 2015 JGR ES 120 homogeneous model), I think the model would have produced results that matched the alpine example more closely. The minimum I would like to see is a discussion of why these particular boundary conditions were chosen and how the authors think that the boundary conditions impact on the model.

R: We agree with the reviewer; the proposed boundary conditions are an oversimplification of natural mountain building conditions and the simulated relief is similar to some extent to an island rising out of the sea. This choice has been driven by two reasons. First, this type of conditions has been extensively used to study mountain dynamic in both experimental and numerical models (Bonnet & Crave, 2003, Bonnet 2005, Graveleau 2008, Castelltort et al., 2013 to cite a few). Second, considering that the boundary conditions remain fixed we do not need to specify any periodic or symmetric boundaries when performing the LEC calculation, it greatly increases the Dijkstra's algorithm runtime. However, and following reviewer's comment, we have added a new section in the discussion (section 4.1) where we present a new scenario similar to the homogeneous model proposed in Roy et al. 2015 (we have also added 2 new figures Figs 6 & 7). In this section, we discuss the impact of boundary conditions and species niche width on the resulting LEC distribution.

SC2: I'm concerned about some of the geomorphic statements in the paper. This model is completely erosional, yet it is stated that the topography on the leeward side is constructional (page 9). Constructional topography is formed by block movement, for example, a range is uplifted along a thrust fault and a river may then be constrained to flow along the valley formed by that block uplift. An example is Central Otago, New Zealand where rivers flow parallel to the main ranges. The relief associated with the topography in this region is a result of the block movements along the faults. In this model, the whole block is being uplifted as one and any relief that arises is because of differential erosion.

R: We agree with the reviewer and the paragraph on page 9-10 has been modified to reflect the correction proposed by the reviewer.

SC3: I think the differences between the two models presented here are overstated. Page 10, line 2 "...has profound effects on the geomorphic expressions of the simulated landscape". The two landscapes are different in detail, but they are both dendritic, erosional landscapes.

R: Roe et al. (2003) have shown that orographic feedback is an important component of the climate-erosion-uplift system that ultimately controls the topographic (and to some degree geological) evolution of mountain ranges. In our experiment, we see that this feedback induces strong variations in slope/gradient and chi values across the landscape and drainage divide. Yet we don't want to overstate the differences between these two models and based on reviewer comment, we have added in the 'Experimental setting' section a sentence stating that the two simulated landscapes are both dendritic end erosional. We have also modified the section page 10 to make sure we do not exaggerate the differences between the simulations.

SC4: These last two points raise an interesting question, which may be beyond the scope of this study, is there a significant difference in the LEC for erosional landscapes (e.g., this study, Bertuzzo et al 2016) and constructional landscapes (e.g., Central Otago – Craw et al. 2016, Nat Geosci, 9; or landscapes further north in New Zealand – Craw et al., 2019, Geomorphology, 336)? Inclusion in the discussion that this model only addresses one type of landscape (erosional) and how other landscapes might differ would add weight to the manuscript.

R: We agree with this suggestion and are now mentioning the limitation of our current simulation (erosional only) in the new section 4.1. (see our response to SC1). We have also added the reference to Craw et al., 2019 in this section.

SC5: I'd like to see more like-for-like plots. Figures 2 and 4 are just different enough that it is hard to see the differences between the uniform rain and the orographic rain

models. Similarly, an equivalent to figure 5 for the uniform rain model might be useful.

R: We think that we have provided equivalent plots between the 2 simulations. As an example, the differences presented between the LEC distribution obtained at steady state for the uniform rainfall (Fig. 2a) and for the orographic (Fig. 4d) scenarios are like-for-like plots. Yet, based on the reviewer comments, we have added line 3 on page 11 a reference to the aforementioned figures to ease the comparisons between the 2 simulations. We have decided not to add an equivalent to Fig. 5 for the uniform rain. In comparison to the orographic case, we do not observe any substantial differences between the 2 catchments facing a similar drainage divide. The result is similar to Fig. 4E in Bertuzzo et al. 2016 with the blue and red dots at the same elevation range. When tectonic stops (the relaxation phase), we observe a similar trend as the one observed in Fig. 5 with an overall drop in LEC values for the considered catchment (similar to what is described in Fig. 3c).

SC6: Figure 2: The caption to (c) states "...generally low LEC for valleys and mountain tops and high LEC on the most elevated flanks of the mountain ranges." I don't think this reflects what is being shown. I see low LEC for valleys and minor peaks lower on the ridges, some higher LEC on ridge flanks but the highest LEC along the highest elevations and along river valleys at 5.5Ma. This is quite different to what is shown in Bertuzzo et al (2016) where the highest elevations are very low LEC. I suspect a plot of LEC vs elevation for this model would not be a bell curve centred on mid-elevations but would be dominated by higher elevations – as is my reading of figure 3A. I think this is a function of the landscape model chosen, as outlined above in the first comment. This may also be being accentuated by the colour map chosen. I find figure 2C hard to reconcile with the text, figure 3 and Bertuzzo et al. (2016). Going from red to blue with a small band of white between them means a lot of the grid appears to be very high LEC when maybe it isn't. I would recommend a different colour map to be used.

R: We have changed the caption in Figure 2c as proposed by the reviewer. The distribution of LEC vs elevation is provided in Fig. 3a and c and shows high elevation

distribution not at the highest elevation but on the most elevated flank as described in the text. The LEC distribution has been obtained following a similar approach as the one proposed in Bertuzzo et al. 2016. In comparisons to Fig 4F and top Fig. S4 in Bertuzzo et al. (2016), our simulated landscape produces higher LEC at higher elevations, but this is likely related to the chosen boundary conditions and the symmetric nature of our model. As shown in section 4.1, with asymmetric boundary conditions similar to the homogeneous model proposed in Roy et al. 2015 the peak in LEC migrates toward mid-elevations (Fig. 6 & 7). We also found similar observation when we increased the species niche width in the LEC calculation (Fig. 6 & 7).

Technical corrections:

TC1: Page 1, Line 17: replace 'This competition converts...' with "These competing processes convert...

R: corrected

TC2: Line 20: timescales

R: corrected

TC3: Line 24: distances

R: corrected

TC4: Page 2, Line 3: change geological to tectonic

R: corrected

TC5: Line 8: reword 'highest frequencies for mid-elevations'

R: corrected

TC6: Line 8: over geological time? Implies a very long time. Reword to entire orogenic system

R: corrected

TC8: Line 19: as outlined above, I don't think the geomorphic processes are changing, there is a difference in rates due to the change in the distribution of the precipitation driving the model, but it is still based on a stream power/hillslope diffusion model, no change in process.

R: We agree and line 19 now states that in our model changes in LEC is the result of climate, tectonic and geomorphic processes.

TC9: Line 31: allows us to

R: corrected

TC10: Page 4, 1st line of caption: check tense here, are run or were ran

R: corrected

TC11: Page 5, Line 16: spelling - cycle

R: corrected

TC12: Line 17: check tense here, are run or were ran

R: corrected

TC13: Line 25: is this a series of simulations or two simulations?

R: corrected

TC14: Lines 32-33: references for this assumption

R: references (Mokany et al. (2012) and Steinbauer et al. (2016) have been added.

TC15: Page 7, lines 20-21: '... model predicts that, on average, higher LEC are found at intermediate elevations.' None of the figures really show this, Figure 3a at 5Ma has a broad peak across most of the elevation range and (c) has a peak at very low elevations.

R: the text has been modified based on the reviewer's comment.

**ESurfD**
TC17: Lines 29-30: again I don't see an obvious peak at mid-elevations, rather a broad peak across most elevations.

R: here as well we have modified the text and a reference to section 4.1 is also provided.

TC18: Page 10, 4th line of caption: defined not defines

R: corrected

TC19: Page 12, line 3: ...the offspring can come from the...

R: corrected

TC20: Line 8: allows us to model

R: corrected

TC21: Line 9: ... a uniform

R: corrected

TC22: Page 13, line 9: why is this? Because these slopes aren't as steep therefore allowing for more distance with change in elevation?

R: indeed, windward facing regions have lower slope and therefore higher surface area than leeward regions. Because the LEC calculation computes the cost as a measure of elevation change with distance, the connectivity will be higher for lower elevational change (windward region).

TC23: Page 15, 1st line of figure caption: replace in at end of line with of

R: corrected

TC24: Figure 3: Plots (a) and (c) appear to have a moving horizontal scale, that is zmin and zmax change with time. State in the caption for clarity.

R: corrected

TC25: Figure 4: (d) is a map of chi over the whole region. Willett et al. (2014) show chi calculated along streams not the whole region. I think it would be much easier to see what the authors are trying to show here if this plot was similar to Willett et al Figure 3 or 4. The scale bar being pushed toward lower values also makes it difficult.

R: Our chi values are calculated along stream as in Willett et al. (2014), but due to the resolution and the size of the shown region it looks like a map. We have provided a supplementary figure SFig1 attached to our reply that shows in the left panel (SFig1a) the Chi values as in Figure 4e in the manuscript as well as a zoom region R1 showing Chi along the stream (SFig1b) equivalent to Willett et al. (2014) figures.

TC26: Figure 5b 5.5My. Given that the area shown in (a) is very small compared to the entire domain, I'm surprised by the distribution of the grey dots beneath those of catchment C1 and C2. Do these represent the whole domain? Shouldn't there be many more grey dots around the elevation range of C1 and C2? This plot suggests that the chosen region is unusual compared to the rest of the domain, but this is not evident in figure 5b.

R: we agree with reviewer and have changed the label in the figure. The colour dots correspond to the points shown in Fig. 5a and the grey dots to all the points present in catchment C1 and C2.

TC27: Figure 8: How is an isolated region defined? The text only specifies low LEC and the figure caption lower connectivities. The final two sentences of the caption appear to refer to (a) not (b). The third plot of 8b doesn't look as asymmetric as the second plot of figure 7b.

R: in Fig. 8 isolated regions are defined based on a normalised LEC below 20%. We have added this information in the text and the caption. The plots are indeed different because they do not correspond to the same time interval (5.0 Myr for Fig. 7b and 5.25 for 8b). During the relaxation phase as discussed in section 3.1 and also in Fig. 3, there is a fast decrease in LEC in the highest regions of the domain.

TC28: There are some inconsistencies between the figures. (1) The region of 4d shown in figures 7 and 8 don't appear to be the same. The third plot of 8b doesn't look as asymmetric as the second plot of figure 7b. (2) Scale bars for LEC, figure 8 has more white around the median value of 0.5 than in other figures. This makes comparisons difficult.

R: the reviewer was right, the plots on Fig. 8b were located south of the region C from Fig. 4d. We have created new plots centred on the region C and we have also changed the LEC colour bar in this figure to make it consistent with the other figures in the manuscript.

Please also note the supplement to this comment:
https://www.earth-surf-dynam-discuss.net/esurf-2019-32/esurf-2019-32-SC1-supplement.pdf

[Figure]

[Figure]

**Fig. 1.**

**Supplement:**

[revised manuscript text omitted]

---

## Referee Comment (RC2) · Anonymous Referee #2 · 29 Jul 2019

In this manuscript, the authors combined geomorphological, meteorological, and ecological models to explore how biodiversity patterns evolve over a geological timescale. The work appeared to me to be an intensive, computational undertaking for which the authors should be applauded. The authors adopted the so-called landscape elevational connectivity (LEC) as a simplified proxy of biodiversity and analyzed how the LEC's spatial distribution changes over a geological timescale. They then extracted a number of insights from these LEC patterns. The manuscript is generally well-written and reads well. The topic should appeal to the ESD audience. Below I discuss a few reservations and make some suggestions which I hope will help improve the manuscript.

The extent of validity of LEC as a measure of biodiversity in mountainous regions. At many places throughout the manuscript, the authors state that LEC can explain "to

the first order" the biodiversity found in mountainous regions. This claim seems to be based on Fig. 6. But the claim will only be valid if the y-axis of Fig. 6 is EMPIRICAL biodiversity; as it is, it is from model results. While this type of biodiversity metacommunity model has been shown to produce realistic biodiversity patterns for a range of ecological systems, I am not sure if it has been done for mountainous regions with such a wide range of environmental conditions and niches. A citation or two that show this is the case will strengthen the above claim. Otherwise, a caveat/caution should be placed earlier in the manuscript.

The qualifier "neutral" is not necessary nor accurate. In your model, individuals of different species have different optimal elevations (z_opt) and therefore are not equivalent. Consequently, some readers may be confused by calling this model "neutral." Indeed, the reference to the neutral model is not needed nor helpful here, in my opinion.

Is there a more intuitive way to explain/understand chi (Eq. 4)? I found it a bit difficult to interpret and appreciate results associated with this quantity.

Other minor comments:

Fig 1's caption: "Two scenarios are ran" -> "Two scenarios are run"

P7, L3: "In this first example" -> "In this first set of results"? For me, this is not an example, but an experimental setting or something along that line.

P8, L2: "inhomogeneous" -> "heterogenous"

Fig 5's caption, 4th line: "are defines" -> "are defined"

P12, L13-14: I suggest changing "In addition and for simplicity as we assume a neutral approach, the parameter....1" to "The parameter fmax does not affect the system dynamics and is, without loss of generality, set to 1."

P14, L25: "Orographic precipitation fosters faster isolation than the uniform precipitation"–I must admit that I had a hard time finding the figure that supports this

statement.

---

## Author Response (AR2)

**REVIEWER 1**

First, we would like to thank the reviewer, Phaedra Upton, for her comments. Below we provide a point by point response to these comments. We have also attached the updated manuscript.

**Specific comments:**

**SC1:** *I don't think the boundary conditions chosen for the landscape model do justice to the study. The model used is a block of rock uplifting at a fixed rate with potential outlets on all sides. The resulting landscape is predictably a mountain with four low angled ridges ending at the four corners of the model. It best reflects an island rising out of the sea. The paper that proposed geomorphic controls on elevational gradients of species richness (Bertuzzo et al. 2016) uses a region of the Swiss Alps. With slightly different boundary conditions, for example, outlets along only one boundary (e.g., Roy et al. 2015 JGR ES 120 homogeneous model), I think the model would have produced results that matched the alpine example more closely. The minimum I would like to see is a discussion of why these particular boundary conditions were chosen and how the authors think that the boundary conditions impact on the model.*

**R:** We agree with the reviewer; the proposed boundary conditions are an oversimplification of natural mountain building conditions and the simulated relief is similar to some extent to an island rising out of the sea. This choice has been driven by two reasons. First, this type of conditions has been extensively used to study mountain dynamic in both experimental and numerical models (Bonnet & Crave, 2003, Bonnet 2005, Graveleau 2008, Castelltort et al., 2013 to cite a few). Second, considering that the boundary conditions remain fixed we do not need to specify any periodic or symmetric boundaries when performing the LEC calculation, it greatly increases the Dijkstra's algorithm runtime. However, and following reviewer's comment, we have added a new section in the discussion (section 4.1) where we present a new scenario similar to the homogeneous model proposed in Roy et al. 2015 (we have also added 2 new figures Figs 6 & 7). In this section, we discuss the impact of boundary conditions and species niche width on the resulting LEC distribution.

**SC2:** *I'm concerned about some of the geomorphic statements in the paper. This model is completely erosional, yet it is stated that the topography on the leeward side is constructional (page 9). Constructional topography is formed by block movement, for example, a range is uplifted along a thrust fault and a river may then be constrained to flow along the valley formed by that block uplift. An example is Central Otago, New Zealand where rivers flow parallel to the main ranges. The relief associated with the topography in this region is a result of the block movements along the faults. In this model, the whole block is being uplifted as one and any relief that arises is because of differential erosion.*

**R:** We agree with the reviewer and the paragraph on page 9-10 has been modified to reflect the correction proposed by the reviewer.

**SC3:** *I think the differences between the two models presented here are overstated. Page 10, line 2 "...has profound effects on the geomorphic expressions of the simulated landscape". The two landscapes are different in detail, but they are both dendritic, erosional landscapes.*

**R:** Roe et al. (2003) have shown that orographic feedback is an important component of the climate-erosion-uplift system that ultimately controls the topographic (and to some degree geological) evolution of mountain ranges. In our experiment, we see that this feedback induces strong variations in slope/gradient and chi values across the landscape and drainage divide. Yet we don't want to overstate the differences between these two models and based on reviewer comment, we have added in the `Experimental setting` section a sentence stating that the two simulated landscapes are both dendritic end erosional. We have also modified the section page 10 to make sure we do not exaggerate the differences between the simulations.

**SC4:** *These last two points raise an interesting question, which may be beyond the scope of this study, is there a significant difference in the LEC for erosional landscapes (e.g., this study, Bertuzzo*

*et al 2016) and constructional landscapes (e.g., Central Otago – Craw et al. 2016, Nat Geosci, 9; or landscapes further north in New Zealand – Craw et al., 2019, Geomorphology, 336)? Inclusion in the discussion that this model only addresses one type of landscape (erosional) and how other landscapes might differ would add weight to the manuscript.*

**R:** We agree with this suggestion and are now mentioning the limitation of our current simulation (erosional only) in the new section 4.1. (see our response to SC1). We have also added the reference to Craw et al., 2019 in this section.

**SC5:** *I'd like to see more like-for-like plots. Figures 2 and 4 are just different enough that it is hard to see the differences between the uniform rain and the orographic rain models. Similarly, an equivalent to figure 5 for the uniform rain model might be useful.*

**R:** We think that we have provided equivalent plots between the 2 simulations. As an example, the differences presented between the LEC distribution obtained at steady state for the uniform rainfall (Fig. 2a) and for the orographic (Fig. 4d) scenarios are like-for-like plots. Yet, based on the reviewer comments, we have added line 3 on page 11 a reference to the aforementioned figures to ease the comparisons between the 2 simulations. We have decided not to add an equivalent to Fig. 5 for the uniform rain. In comparison to the orographic case, we do not observe any substantial differences between the 2 catchments facing a similar drainage divide. The result is similar to Fig. 4E in Bertuzzo et al. 2016 with the blue and red dots at the same elevation range. When tectonic stops (the relaxation phase), we observe a similar trend as the one observed in Fig. 5 with an overall drop in LEC values for the considered catchment (similar to what is described in Fig. 3c).

**SC6:** *Figure 2: The caption to (c) states "...generally low LEC for valleys and mountain tops and high LEC on the most elevated flanks of the mountain ranges." I don't think this reflects what is being shown. I see low LEC for valleys and minor peaks lower on the ridges, some higher LEC on ridge flanks but the highest LEC along the highest elevations and along river valleys at 5.5Ma. This is quite different to what is shown in Bertuzzo et al (2016) where the highest elevations are very low LEC. I suspect a plot of LEC vs elevation for this model would not be a bell curve centred on mid-elevations but would be dominated by higher elevations – as is my reading of figure 3A. I think this is a function of the landscape model chosen, as outlined above in the first comment. This may also be being accentuated by the colour map chosen. I find figure 2C hard to reconcile with the text, figure 3 and Bertuzzo et al. (2016). Going from red to blue with a small band of white between them means a lot of the grid appears to be very high LEC when maybe it isn't. I would recommend a different colour map to be used.*

**R:** We have changed the caption in Figure 2c as proposed by the reviewer. The distribution of LEC vs elevation is provided in Fig. 3a and c and shows high elevation distribution not at the highest elevation but on the most elevated flank as described in the text. The LEC distribution has been obtained following a similar approach as the one proposed in Bertuzzo et al. 2016. In comparisons to Fig 4F and top Fig. S4 in Bertuzzo et al. (2016), our simulated landscape produces higher LEC at higher elevations, but this is likely related to the chosen boundary conditions and the symmetric nature of our model. As shown in section 4.1, with asymmetric boundary conditions similar to the homogeneous model proposed in Roy et al. 2015 the peak in LEC migrates toward mid-elevations (Fig. 6 & 7). We also found similar observation when we increased the species niche width in the LEC calculation (Fig. 6 & 7).

**Technical corrections:**

**TC1:** *Page 1, Line 17: replace 'This competition converts...' with "These competing processes convert...*
**R:** corrected

**TC2:** *Line 20: timescales*
**R:** corrected

**TC3:** *Line 24: distances*
**R:** corrected

**TC4:** *Page 2, Line 3: change geological to tectonic*
**R:** corrected

**TC5:** *Line 8: reword 'highest frequencies for mid-elevations'*
**R:** corrected

**TC6:** *Line 8: over geological time? Implies a very long time. Reword to entire orogenic system*
**R:** corrected

**TC8:** *Line 19: as outlined above, I don't think the geomorphic processes are changing, there is a difference in rates due to the change in the distribution of the precipitation driving the model, but it is still based on a stream power/hillslope diffusion model, no change in process.*
**R:** We agree and line 19 now states that in our model changes in LEC is the result of climate, tectonic and geomorphic processes.

**TC9:** *Line 31: allows us to*
**R:** corrected

**TC10:** *Page 4, 1st line of caption: check tense here, are run or were ran*
**R:** corrected

**TC11:** *Page 5, Line 16: spelling - cycle*
**R:** corrected

**TC12:** *Line 17: check tense here, are run or were ran*
**R:** corrected

**TC13:** *Line 25: is this a series of simulations or two simulations?*
**R:** corrected

**TC14:** *Lines 32-33: references for this assumption*
**R:** references (Mokany et al. (2012) and Steinbauer et al. (2016) have been added.

**TC15:** *Page 7, lines 20-21: '... model predicts that, on average, higher LEC are found at intermediate elevations.' None of the figures really show this, Figure 3a at 5Ma has a broad peak across most of the elevation range and (c) has a peak at very low elevations.*
**R:** the text has been modified based on the reviewer's comment.

**TC17:** *Lines 29-30: again I don't see an obvious peak at mid-elevations, rather a broad peak across most elevations.*
**R:** here as well we have modified the text and a reference to section 4.1 is also provided.

**TC18:** *Page 10, 4th line of caption: defined not defines*
**R:** corrected

**TC19:** *Page 12, line 3: ...the offspring can come from the...*
**R:** corrected

**TC20:** *Line 8: allows us to model*
**R:** corrected

**TC21:** *Line 9: ... a uniform*
**R:** corrected

**TC22:** *Page 13, line 9: why is this? Because these slopes aren't as steep therefore allowing for more distance with change in elevation?*
**R:** indeed, windward facing regions have lower slope and therefore higher surface area than leeward regions. Because the LEC calculation computes the cost as a measure of elevation change with distance, the connectivity will be higher for lower elevational change (windward region).

**TC23:** *Page 15, 1st line of figure caption: replace in at end of line with of*
**R:** corrected

**TC24:** *Figure 3: Plots (a) and (c) appear to have a moving horizontal scale, that is zmin and zmax change with time. State in the caption for clarity.*
**R:** corrected

**TC25:** *Figure 4: (d) is a map of chi over the whole region. Willett et al. (2014) show chi calculated along streams not the whole region. I think it would be much easier to see what the authors are trying to show here if this plot was similar to Willett et al Figure 3 or 4. The scale bar being pushed toward lower values also makes it difficult.*
**R:** Our chi values are calculated along stream as in Willett et al. (2014), but due to the resolution and the size of the shown region it looks like a map. We have provided a supplementary figure (below) that shows in the left panel (a) the Chi values as in Figure 4e in the manuscript as well as a zoom region R1 showing Chi along the stream (b) equivalent to Willett et al. (2014) figures.

[Figure]

**TC26:** *Figure 5b 5.5My. Given that the area shown in (a) is very small compared to the entire domain, I'm surprised by the distribution of the grey dots beneath those of catchment C1 and C2. Do these represent the whole domain? Shouldn't there be many more grey dots around the elevation range of C1 and C2? This plot suggests that the chosen region is unusual compared to the rest of the domain, but this is not evident in figure 5b.*
**R:** we agree with reviewer and have changed the label in the figure. The colour dots correspond to the points shown in Fig. 5a and the grey dots to all the points present in catchment C1 and C2.

**TC27:** *Figure 8: How is an isolated region defined? The text only specifies low LEC and the figure caption lower connectivities. The final two sentences of the caption appear to refer to (a) not (b). The third plot of 8b doesn't look as asymmetric as the second plot of figure 7b.*
**R:** in Fig. 8 isolated regions are defined based on a normalised LEC below 20%. We have added this information in the text and the caption. The plots are indeed different because they do not correspond to the same time interval (5.0 Myr for Fig. 7b and 5.25 for 8b). During the relaxation phase as discussed in section 3.1 and also in Fig. 3, there is a fast decrease in LEC in the highest regions of the domain.

**TC28:** *There are some inconsistencies between the figures. (1) The region of 4d shown in figures 7 and 8 don't appear to be the same. The third plot of 8b doesn't look as asymmetric as the second plot of figure 7b. (2) Scale bars for LEC, figure 8 has more white around the median value of 0.5 than in other figures. This makes comparisons difficult.*
**R:** the reviewer was right, the plots on Fig. 8b were located south of the region C from Fig. 4d. We have created new plots centred on the region C and we have also changed the LEC colour bar in this figure to make it consistent with the other figures in the manuscript.

**REVIEWER 2**

We would like to thank the anonymous reviewer for his comments. Below we provide a point by point response. We have also attached the updated manuscript based on the two reviews.

**Main comments:**

**MC1:** *The extent of validity of LEC as a measure of biodiversity in mountainous regions. At many places throughout the manuscript, the authors state that LEC can explain "to the first order" the biodiversity found in mountainous regions. This claim seems to be based on Fig. 6. But the claim will only be valid if the y-axis of Fig. 6 is EMPIRICAL biodiversity; as it is, it is from model results. While this type of biodiversity metacommunity model has been shown to produce realistic biodiversity patterns for a range of ecological systems, I am not sure if it has been done for mountainous regions with such a wide range of environmental conditions and niches. A citation or two that show this is the case will strengthen the above claim. Otherwise, a caveat/caution should be placed earlier in the manuscript.*

**R:** The reviewer is right when mentioning that the y-axis of Fig. 6 is based on model results and it is fair to acknowledge that a recent study from Liu et al. (2018) investigated the relationship between species richness and elevation on ant community within the Hengduan Mountains region and did not find a similar relationship as the one from Bertuzzo et al. (2016). The authors agree that environmental gradients dominate variation in both alpha and beta diversity but in their case ant alpha diversity strongly declines with elevation. However, several empirical observations often show a hump-shaped rather that monotically decreasing pattern such as in Lomolino (2001), Rahbek (1995, 2005), McCain & Grytnes (2010) or Kessler et al. (2011). As suggested by the reviewer we have added references to these studies. We believe it highlights the power of a pluralistic approach integrating field surveys with conceptual, statistical, and theoretical frameworks to understand the drivers of species distribution patterns. Future research bridging the gap between theory and the real-world systems will enhance our understanding of the mechanisms that govern biodiversity patterns.

**MC2:** *The qualifier "neutral" is not necessary nor accurate. In your model, individuals of different species have different optimal elevations (z_opt) and therefore are not equivalent. Consequently, some readers may be confused by calling this model "neutral." Indeed, the reference to the neutral model is not needed nor helpful here, in my opinion.*

**R:** We agree with the reviewer and have removed the qualifier neutral in the text when referring to the metacommunity model. We have also removed the reference to Hubbell 2001.

**MC3:** *Is there a more intuitive way to explain/understand chi (Eq. 4)? I found it a bit difficult to interpret and appreciate results associated with this quantity.*

**R:** Chi analysis is a method of extracting information from channel profiles that attempts to compare channels with different discharges (Perron & Royden, 2013). The longitudinal coordinate chi has dimensions of length and is linearly related to the elevation z(x). Therefore, if a channel incises based on the stream power incision model like in our landscape evolution model, then its profile should be linear on a plot of elevation against chi. As well as providing a method to test whether channel profiles obey common incision models, chi-plots could also be used in the field to provide means of testing the appropriate m/n for a channel (Mudd et al., 2014). In the paper we have added first a reference to Mudd et al. (2014) as well as additional explanation in regard to the use of chi parameter.

**C1:** *Fig 1's caption: "Two scenarios are ran" -> "Two scenarios are run"*
**R:** corrected.

**C2:** *P7, L3: "In this first example" -> "In this first set of results"? For me, this is not an example, but an experimental setting or something along that line.*
**R:** corrected.

**C3:** *P8, L2: "inhomogeneous" -> "heterogenous"*
**R:** corrected.

**C4:** *Fig 5's caption, 4th line: "are defines" -> "are defined"*
**R:** corrected.

**C5:** *P12, L13-14: I suggest changing "In addition and for simplicity as we assume a neutral approach, the parameter. . ..1" to "The parameter fmax does not affect the system dynamics and is, without loss of generality, set to 1."*
**R:** corrected.

**C6:** *P14, L25: "Orographic precipitation fosters faster isolation than the uniform precipitation"–I must admit that I had a hard time finding the figure that supports this*
**R:** We have added in the revised manuscript the associated figure panel as well as the lines within the plots that supports this statement (red lines in Fig. 10a).

[revised manuscript text omitted]